# Sequential membrane remodeling by cholesterol distinctly modulates HCN channels in naïve and neuropathic DRG neurons

Lucas J. Handlin[1], Clémence Gieré[2], Nicolas L.A. Dumaire[2], Lyuba Salih[2], Aubin Moutal[2], and Gucan Dai[1]

Cholesterol, abundantly present in distinct plasma membrane pools, is a critical modulator of ion channel function, including hyperpolarization-activated cyclic nucleotide-gated (HCN) channels that regulate the excitability of dorsal root ganglion (DRG) nociceptor neurons. Depletion of membrane cholesterol potentiated HCN channel opening and accelerated activation kinetics, whereas cholesterol supplementation reduced channel opening and slowed activation kinetics. However, the relative contributions of cholesterol that organizes ordered membrane domains (OMDs) versus freely accessible cholesterol pools to HCN channel modulation remain unknown. Using fluorescence lifetime imaging microscopy, FRET and fluorescence anisotropy techniques, we examined how supplementing cholesterol alters plasma membrane properties and HCN gating in nociceptor DRG neurons. We uncovered a process of sequential, stepwise membrane remodeling: an initial phase with OMD expansion and a rapid rise in free cholesterol, followed by continued accumulation of free cholesterol without further OMD expansion. Notably, the slope factor of the HCN G-V relationship is sensitive to OMD expansion but remains unaffected by changes in free cholesterol. Other gating parameters, including open probability and activation kinetics, were affected by elevating free cholesterol. In a rat model of nerve injury, where DRG neurons exhibit reduced free cholesterol levels and smaller OMDs, HCN channel modulation by cholesterol involves contributions from both OMD expansion and free cholesterol accumulation. In contrast, in naïve DRG neurons—characterized by high cholesterol and large OMDs—modulation occurs mostly via increased free cholesterol. These findings provide mechanistic insights into cholesterol-dependent modulation of ion channels and its role in neuropathic pain.

## Introduction

Cholesterol is a fundamental component of the eukaryotic plasma membrane (PM), accounting for about 30–50 mol% of total lipids in mammalian cells and exerting profound effects on membrane structure and function (Veatch and Keller, 2002; Simons and Ehehalt, 2002; Liu et al., 2017). Its rigid sterol ring structure embeds between the acyl chains of membrane lipids, reducing their motional freedom and promoting lipid packing, thereby increasing membrane order and thickness while decreasing permeability to small polar molecules (Corvera et al., 1992; Subczynski et al., 2017). Beyond these physical effects, cholesterol contributes to the lateral heterogeneity of the membrane by driving the formation of lipid-ordered membrane domains (OMDs), which serve as organizing platforms for proteins and signaling assemblies (Simons and Ehehalt, 2002; Lingwood and Simons, 2010; Leventhal and Lyman, 2023). The asymmetric distribution of cholesterol between the inner and outer leaflets, as well as its dynamic exchange between membrane pools, adds further complexity to its regulatory functions

(Das et al., 2014; Liu et al., 2017). Given its central role in shaping membrane architecture and influencing protein–lipid interactions, cholesterol has emerged as a key regulator of cellular excitability and signal transduction (Kinnebrew et al., 2019; Wang et al., 2021; Handlin et al., 2024; Robinson et al., 2019).

Cholesterol-mediated modulation of ion channel function remains an area of intense investigation, with longstanding debates about whether its effects arise from direct molecular interactions or from alterations in membrane organization (Levitan et al., 2014; Beverley and Levitan, 2024; Furst and D'Avanzo, 2015; Veatch and Keller, 2002). Emerging evidence suggests that cholesterol-rich, lipid-ordered membrane compartments—also known as OMDs—play a crucial role in organizing membrane proteins and signaling complexes, thereby influencing the functional properties of ion channels (Lingwood and Simons, 2010; Leventhal and Lyman, 2023; Hilgemann et al., 2018; Handlin et al., 2024; Veatch and Keller, 2002). Among these, hyperpolarization-activated cyclic nucleotide-gated (HCN)

[1]Edward A. Doisy Department of Biochemistry and Molecular Biology, Saint Louis University School of Medicine, St. Louis, MO, USA; [2]Department of Pharmacology and Physiology, Saint Louis University School of Medicine, St. Louis, MO, USA.

Correspondence to Gucan Dai: gucan.dai@health.slu.edu.

channels regulate neuronal excitability through generating "$I_h$ current," which is mostly sodium inward current in physiological conditions that promotes membrane depolarization (DiFrancesco, 1986; Saponaro et al., 2021a; Zagotta et al., 2003; Craven and Zagotta, 2006). For somatosensory neurons, small-diameter nociceptor dorsal root ganglion (DRG) neurons predominantly express the HCN2 isoform and exhibit relatively slow HCN currents that are highly sensitive to cAMP (Momin et al., 2008; Emery et al., 2011; Handlin et al., 2024; Saponaro et al., 2021b). In contrast, larger DRG neurons display faster HCN currents and primarily express the HCN1 isoform, which shows limited sensitivity to cAMP (Momin et al., 2008; Emery et al., 2011). Aberrant upregulation or altered gating of HCN channels in sensory neurons has been linked to increased excitability and spontaneous firing, hallmarks of chronic pain states (Emery et al., 2011; Chaplan et al., 2003; Young et al., 2014). Furthermore, HCN2 and HCN4 isoforms are thought to reside in OMDs, whereas HCN1 is predominantly localized to disordered membrane regions (Handlin and Dai, 2023; Handlin et al., 2024). Thus, the OMD-mediated modulation of HCN channels, particularly HCN2 channels in small nociceptor DRG neurons, has emerged as a mechanistic mechanism underlying the pathophysiology of neuropathic pain (Emery et al., 2011; Emery et al., 2012; Lainez et al., 2019; Furst and D'Avanzo, 2015; Handlin and Dai, 2023; Handlin et al., 2024).

The voltage sensor activity and function of HCN channels are modulated by their localization within OMDs, suggesting a lipid-dependent mechanism of regulation (Handlin and Dai, 2023; Handlin et al., 2024). We recently uncovered that disruption of OMDs through cholesterol extraction dramatically alters native HCN currents, pointing to OMDs as key modulators of HCN channel activity in the context of neuropathic pain (Handlin et al., 2024). Cholesterol extraction and supplementation modulate action potential firing in small nociceptor DRG neurons, in part by altering OMD size and consequently affecting HCN channel gating (Handlin et al., 2024). Understanding how cholesterol and OMDs regulate HCN channels could reveal novel therapeutic strategies for targeting pathological hyperexcitability in neuropathic pain.

This study provides evidence that cholesterol dynamically regulates HCN channel gating through a dual mechanism: by modifying OMDs and by altering free, accessible cholesterol levels in the inner leaflet of the PM. By integrating fluorescence lifetime imaging (FLIM), FRET, genetically encoded cholesterol sensors, and homo-FRET–based anisotropy imaging, we reveal a sequential response to cholesterol enrichment at the PM: an initial OMD expansion, followed by a sustained increase in inner and outer leaflet cholesterol. This distinction allows us to temporally separate the impact of OMDs from that of cholesterol accumulation, addressing a persistent challenge in membrane biophysics. Our findings have direct implications for the pathophysiology of neuropathic pain. In a nerve injury model, we show that DRG neurons exhibit smaller OMDs and reduced inner leaflet cholesterol. *In vivo*, cholesterol depletion induces mechanical hypersensitivity in naïve rats, while cholesterol supplementation alleviates mechanical hypersensitivity in a model of chronic neuropathic pain. These changes correlate with altered HCN channel gating, providing a mechanistic link between lipid remodeling and hyperexcitability in chronic pain states. By elucidating how cholesterol dynamically regulates membrane properties, this study opens new avenues for investigating cholesterol-mediated modulation of ion channels in excitable cells.

## Materials and methods

### Cell culture and molecular reagents

The tsA-201 cell line (RRID: CVCL_2737), a derivative of human embryonic kidney (HEK293) cells, was obtained from Sigma-Aldrich (catalog #96121229). The identity of the cell line was authenticated by Sigma using short tandem repeat profiling. Cells were maintained in Dulbecco's Modified Eagle's Medium (DMEM; Gibco) supplemented with 10% FBS (Gibco) and 1% penicillin-streptomycin (Gibco) under humidified conditions at 37°C with 5% $CO_2$ in tissue culture dishes (CELLTREAT). Transfections were performed at 70–90% confluency using the Lipofectamine 3000 Kit (#L3000008; Invitrogen) following the manufacturer's protocol. Mycoplasma testing was performed using the MycoFluor Mycoplasma Detection Kit (Invitrogen) and tested negative. The tsA-201 cell line was selected because, similar to HEK293T cells, it exhibits high transfection efficiency and robust heterologous expression of proteins. Importantly, tsA-201 cells maintain cholesterol and membrane lipid composition representative of mammalian cells. These properties make tsA-201 cells an ideal heterologous system for examining how changes in membrane cholesterol and OMDs regulate membrane function.

Cholera toxin subunit B (CTxB) conjugated to Alexa Fluor 488 and 647 was obtained from Thermo Fisher Scientific and applied to cultured cells at a concentration of 20 nM for ∼5–10 min before imaging. CTxB binds to GM1-enriched OMDs with a picomolar affinity. The eGFP-GRAM-W plasmid was obtained from Addgene (#211701). The C-terminal mCherry-tagged ostreolysin A (OlyA) was synthesized and purified by GenScript and was applied to cultured cells at a concentration of 1 μM for ∼1 h. Before imaging, OlyA-treated cells were incubated in regular DMEM medium for 30 min to wash away unbound OlyA.

β-cyclodextrin (β-CD), methyl-βCD (mβCD), α-cyclodextrin, and water-soluble cholesterol (WSC) were purchased from Sigma-Aldrich. For same-cell fluorescence and electrophysiology experiments performing cholesterol enrichment, an equal volume of 1 mg/ml WSC in the same external solution was pipetted into the recording chamber, resulting in a final cholesterol concentration of 0.5 mg/ml. For cholesterol-extraction related electrophysiological recordings of HCN currents, an equal volume of 10 mM mβCD in an external solution was pipetted into the recording chamber, generating a final concentration of 5 mM. For fluorescence experiments on tsA cells, 5 mM β-CD was applied using the same method, generating a final concentration of 2.5 mM.

### FLIM-FRET and fluorescence anisotropy

Fluorescence microscopy and FLIM experiments were performed like previous research (Handlin et al., 2024). A laser scanning confocal system with a FastFLIM data acquisition

module (ISS, Inc.) and two hybrid PMT detectors (Hamamatsu) was used for frequency-domain FLIM imaging. A supercontinuum laser (YSL Photonics) provided excitation at specific wavelengths for CFP, GFP, Alexa Fluor 488 (AF-488), and mCherry with consistent laser intensity across samples. Emission was detected using dichroic cubes and filters optimized for CFP/YFP and AF-488/AF-555 FRET pairs. Confocal images (256 × 256 pixels) were acquired using a 100-μm pinhole and pixel dwell times of 0.1–0.4 ms. Image acquisition and processing were performed with VistaVision software (ISS, Inc.), enabling phasor plotting and FLIM analysis (Digman et al., 2008; Handlin et al., 2024; Malacrida et al., 2021; Ranjit et al., 2018). A smoothing filter and intensity-threshold filter were applied to isolate membrane-localized lifetime signals. The phasor FLIM approach distinguished membrane fluorescence from cytosolic and background signals, allowing separation of endocytosed and membrane-localized CTxB. Phase delays (φ) and modulation ratios (m) were determined using Fourier transforms, accounting for the instrument response function calibrated with fluorophores of known lifetimes (Atto 425, rhodamine 110, and rhodamine B) (Malacrida et al., 2021). FLIM-FRET analysis was conducted with the VistaVision FRET trajectory function, optimizing background and donor contributions for accuracy and maintaining background levels below 5% to minimize interference.

Fluorescence anisotropy was used to assess the membrane localization and dynamics of eGFP-glucosyltransferases, Rab-like GTPase activators, and myotubularins (GRAM)-W and mCherry-tagged OlyA proteins. Anisotropy measurements were performed using a polarization module integrated with a confocal microscope, enabling simultaneous acquisition. The system included a half-wave plate and a linear polarizer assembly at the laser entrance, both of which were rotatable with 1° precision over 360° to define the excitation polarization state. Emission fluorescence was separated into parallel ($I_\parallel$) and perpendicular ($I_\perp$) components relative to the excitation polarization using a polarization beam splitter assembly, allowing for precise anisotropy measurements. Cells labeled with membrane-localized GFP- or mCherry-tagged constructs were cultured on glass-bottom petri dishes and maintained in imaging buffer to preserve physiological conditions. Anisotropy was determined using a software package in VistaVision that acquired fluorescence intensity in the two detection channels, with intensity-based background subtraction applied to correct for autofluorescence and cytosolic fluorescence. The fluorescence anisotropy (r) at the PM was calculated using the equation: $r = (I_\parallel - I_\perp)/(I_\parallel + 2{*}I_\perp)$ (Dai et al., 2018; Joshi et al., 2024).

The homo-FRET efficiency of GRAM-W was estimated using fluorescence anisotropy measurements at the PM and in the cytoplasmic region following β-CD treatment. After 5 min of cholesterol extraction with β-CD, GRAM-W exhibited near-complete translocation to the cytosol, where the measured anisotropy of 0.375 was assumed to represent a state minimally affected by homo-FRET and was designated as r′. This assumption is based on the typical cytosolic concentration of constructs following overexpression, which ranges from 1 to 10 μM, corresponding to average intermolecular distances of ~118 nm (for 1 μM) to 55 nm (for 10 μM) in a 3-D space. Given that the Förster distance ($R_0$) for homo-FRET between GFP molecules is ~5 nm, these large intermolecular distances in the cytosol are expected to preclude significant homo-FRET interactions. The homo-FRET efficiency was then calculated using the equation $E_{homo\text{-}FRET} = 1 - r/r'$, where r is the measured anisotropy at the PM and r′ is the cytosolic anisotropy (0.375), serving as the reference value in the absence of homo-FRET (Joshi et al., 2024; van Zanten et al., 2023). The homo-FRET efficiency of OlyA-mCherry was estimated using a similar approach. The anisotropy of OlyA-mCherry in the absence of homo-FRET was measured to be 0.251 using purified OlyA-mCherry dissolved (5 μM) in dPBS.

## Poisson statistics in calculating the probability of CTxB occupancy in response to OMD expansion

We consider the binding affinity $EC_{50}$ of CTxB to the membrane to be 2 nM, and we use C = 20 nM CTxB extracellularly. The percentage of GM1 sites occupied would be $f = C/(C + EC_{50}) = 0.91$, based on the Langmuir isotherm. If we assume the membrane lipid density as 1.54/nm², we can estimate using an example of 30800 GM1 per square micrometer (μm²) or 2 mol% of all lipids within OMD (Fig. 2, B and C; also see 0.5 mol% and 1 mol% of GM1 in the Fig. S1). The stoichiometry between GM1 and CTxB is 5:1.

The mean individual OMD area is calculated using $A_{OMD} = \pi(d/2)^2$, where d is the diameter of OMD, simplified as a circular area. Thus, the mean CTxB occupancy per OMD would be

$$\lambda = 0.91 * 0.0308 * A_{OMD}/5$$

Using Poisson statistics, the probability of a particular number (k) of CTxB per OMD is calculated using the formula

$$P(n = k) = \lambda^k \exp(-\lambda)\big/ k!$$

The probability of more than one CTxB per OMD is calculated as

$$P(k \geq 2) = 1 - \exp(-\lambda) - \lambda \exp(-\lambda)$$

Furthermore, the mean CTxB–CTxB spacing defined by the GM1-imposed surface density is a constant, as it is set by the GM1–CTxB-binding stoichiometry and the GM1 density within OMDs. Consequently, the increase in FRET observed with increasing OMD size arises primarily from the higher probability that two or more CTxB molecules occupy the same OMD, rather than from a change in the local intermolecular spacing of CTxB within OMDs.

## Rat studies and primary DRG neurons

Pathogen-free adult and naïve Sprague-Dawley rats (RRID: RGD_70508; 100–250 g, Envigo) were maintained in a controlled environment with a 12-h light/dark cycle (lights on at 07:00 h) and a regulated temperature of 23 ± 3°C. Food and water were provided *ad libitum*. All experimental procedures were conducted in accordance with National Institutes of Health guidelines and were approved by the Institutional Animal Care and Use Committee of Saint Louis University (protocol number: 3014). Behavioral testing was performed by researchers blinded to experimental conditions.

DRG from the ipsilateral side of nerve injuries were isolated from adult Sprague-Dawley rats. The dissection involved removing dorsal skin and muscle layers, followed by precise cutting of the vertebral bone processes to expose the DRGs. The excised DRGs were trimmed at their roots and enzymatically digested in 3 ml sterile, bicarbonate-free, serum-free DMEM (Catalog# 11965; Thermo Fisher Scientific) supplemented with neutral protease (3.125 mg/ml, Catalog# LS02104; Worthington) and collagenase Type I (5 mg/ml, Catalog# LS004194; Worthington). The tissue was incubated for 45 min at 37°C with gentle agitation. After digestion, the dissociated DRG neurons (~$1.5 \times 10^6$ cells) were collected by centrifugation, washed with DRG culture medium (DMEM supplemented with 1% penicillin-streptomycin sulfate from a 10,000 µg/ml stock, 30 ng/ml nerve growth factor, and 10% FBS), and plated onto 12-mm coverslips coated with poly-D-lysine and laminin.

### Spared nerve injury surgery and von Frey test
Animals were anesthetized with isoflurane (5%) and placed on a heated blanket to preserve body temperature. Anesthesia was maintained (2.5%) throughout the surgery. Areflexia was confirmed before starting the surgery. Chlorhexidine was applied twice on the right leg, and a small incision (2–3 cm) was performed on the thigh. The biceps femoris muscle was bluntly dissected to expose the three terminal branches of the sciatic nerve. The common peroneal and tibial branches were tightly ligated with 4–0 silk and axotomized, leaving the sural branch intact. The muscle was sutured with 5–0 absorbable suture, and the skin was autoclipped. Antibiotic ointment was applied on the skin. Animals were allowed to recover for 7 days before any testing.

Mechanical allodynia was measured by assessing the withdrawal threshold of the paw in response to probing with a series of calibrated fine (von Frey) filaments. Animals were placed in Plexiglas boxes on an elevated mesh screen and were given 15 min to acclimate. Filaments were applied on the plantar surface of both hind paws in ascending forces until a withdrawal reflex or until the cutoff filament of 15 g. The withdrawal threshold was determined by sequentially increasing and decreasing the stimulus strength (the "up and down" method), and data were analyzed with the nonparametric method of Dixon (Chaplan et al., 1994) and expressed as the mean withdrawal threshold.

### Patch-clamp electrophysiology
Whole-cell patch-clamp recordings were conducted using an EPC10 amplifier in conjunction with PATCHMASTER (RRID: SCR_000034) software (HEKA), employing a 5 kHz sampling rate. Borosilicate glass pipettes were fabricated with a P1000 micropipette puller (Sutter Instrument), yielding an initial resistance of 2.5–4 MΩ. A Sutter MP-225A motorized micromanipulator was utilized for precise electrode positioning. Experiments were carried out at room temperature.

To measure HCN currents in DRG neurons, the internal pipette solution was composed of (in mM): 10 NaCl, 137 KCl, 4 Mg-ATP, 10 HEPES, 1 EGTA, and adjusted to pH 7.3 with KOH. The 4 mM Mg-ATP is sufficient to prevent $PI(4,5)P_2$ hydrolysis-

mediated current rundown during the whole-cell patch-clamp experiments (Dai et al., 2016; Pian et al., 2007). The extracellular solution consisted of (in mM): 154 NaCl, 5.6 KCl, 1 MgCl2, 1 CaCl2, 8 HEPES, and 10 D-glucose, with pH set to 7.4 using NaOH. Series resistance was consistently maintained below 10 MΩ, with compensation adjusted between 40 and 70%. The G-V relationship of endogenous HCN currents was derived from instantaneous tail currents measured at −60 mV following test pulses ranging from −40 to −140 mV. Leak-subtracted tail currents at −60 mV were normalized to the maximum tail current amplitude to determine relative conductance ($G/G_{max}$). These values were plotted as a function of the test pulse voltage and fitted to the Boltzmann equation: $G/G_{max} = 1/\{1 + \exp[(V - V_{1/2})/V_S]\}$, where V represents membrane potential, $V_{1/2}$ denotes the voltage for half-maximal activation, and Vs is the slope factor. The slope factor was defined as $V_S = RT/zF$, where R is the gas constant, T is the absolute temperature (297 K), F is the Faraday constant, and z represents the estimated gating charge per channel required for activation. Action potentials were recorded in PATCHMASTER using current-clamp mode rather than voltage clamp (Handlin et al., 2024). Resting membrane potential was set with zero current injection. Absolute current injections were then applied to induce action potentials. DRG neurons showed current-dependent adaptation, with firing rates decreasing as injected current increased.

For whole-cell Na+ channel current recordings, we used a reduced extracellular sodium concentration, as established in previous studies, to decrease the electrochemical driving force and enable more effective voltage clamping of fast Na+ currents (Zhang et al., 2017). The extracellular solution consisted of (in mM): 35 NaCl, 65 choline-Cl, 30 TEA-Cl, 0.1 $CaCl_2$, 5 $MgCl_2$, 0.1 $CdCl_2$, 10 HEPES, and 10 glucose; pH was adjusted to 7.4. The internal pipette solution contained (in mM): 100 cesium methanesulfonate, 40 TEA-Cl, 5 NaCl, 1 $CaCl_2$, 11 EGTA, 10 HEPES, 2 Mg-ATP, and 1 GTP, pH was adjusted to 7.4. A p/−4 leak subtraction method was used to cancel passive capacitive transients. The peak current amplitude at each voltage step was used to construct the G-V relationship for channel activation after correcting for the voltage-dependent driving force. To estimate the driving force at each membrane potential, we determined the reversal potential by linearly extrapolating peak current amplitudes recorded at depolarized voltages to the voltage at which the net current was zero. Steady-state inactivation was assessed using a voltage protocol consisting of pre-pulse steps from −70 to 0 mV, followed by a main test pulse at 0 mV. Peak current amplitudes during the main pulse were normalized to the maximal response and used to estimate postinactivation conductance.

### Statistics and reproducibility
Data parameters are expressed as mean ± SEM from independent cells or patches. All successful single-cell imaging and patch-clamp recordings were included in the analysis; no successful datasets were excluded. Sample size was based on previously published results using patch-clamp recordings of cell lines and primary cells, rat behavior tests, or applying ensemble FRET that generated sufficient statistical power. All experiments were conducted randomly and confirmed to be reproducible.

Key experiments were conducted with at least two independent cell transfections or surgical preparations from animals. The specific number of independent replicates is provided in the figure legends. For single-cell imaging or patch-clamp experiments, statistical significance was assessed using a paired $t$ test for comparisons between before and after treatments. For behavior experiments, two-way ANOVA or Mann–Whitney tests were used. Statistical thresholds were set at *P < 0.05 and **P < 0.01, respectively.

### Online supplemental material

Fig. S1 shows Poisson statistics–based simulations illustrating the increased probability of CTxB binding as OMD size increases. Fig. S2 shows confocal imaging showing the effects of cholesterol extraction or supplementation on the membrane localization of GRAM-W-eGFP. Fig. S3 shows CTxB-based FLIM-FRET responses to cholesterol enrichment in naïve and spared nerve injury (SNI) nociceptor DRG neurons. Fig. S4 shows that WSC treatment does not change the fluorescence lifetime of eGFP-GRAM-W. Fig. S5 shows an increase in current amplitude and acceleration of the HCN channel activation by cholesterol depletion in the presence of a saturating concentration of cAMP. Fig. S6 shows control experiments examining HCN channel gating in naïve DRG nociceptors without WSC treatment, in comparison with conditions with WSC treatment. Fig. S7 shows the effects of cholesterol extraction on the gating of endogenous sodium currents of naïve nociceptor DRG neurons. Table S1 shows the effects on HCN gating parameters by cholesterol supplementation.

## Results

### Cholesterol levels modulate membrane excitability and mechanical pain sensitivity

Cholesterol enrichment reduced the hyperexcitability of SNI nociceptor DRG neurons *in vitro*, consistent with a suppressive effect on HCN channel activity in our experiments (Handlin et al., 2024). We observed considerable effects in the same patch experiment, consistent with our previous findings using β-CD or WSC pretreatments on DRG neurons (Handlin et al., 2024). In non-spontaneously firing naïve DRG neurons, cholesterol extraction induced spontaneous firing (Fig. 1 A), whereas in spontaneously firing neurons, it increased firing frequency (Fig. 1 B). In the same-patch experiment, WSC treatment markedly suppressed action potential firing, as demonstrated by current-clamp recordings with incremental current injections (Fig. 1, C and D). These same-patch treatment data highlight the critical role of cholesterol in regulating the membrane excitability of sensory neurons.

To examine whether cholesterol manipulation influences pain behaviors *in vivo* and to assess its physiological impact, we tested if a local administration of cholesterol-depleting or cholesterol-supplementing agents in naïve or SNI neuropathic pain rats resulted in changes in mechanical hypersensitivity. Intraplantar injection of β-CD, but not its inactive analog α-cyclodextrin, induced significant mechanical hypersensitivity in naïve rats, as evidenced by a time-dependent reduction in

paw withdrawal threshold on the injected paw compared with the contralateral control paw and α-cyclodextrin–injected animals (Fig. 1 E). This effect persisted for 24 h and was quantitatively confirmed (Fig. 1 F), highlighting a pronociceptive effect of acute cholesterol chelation in naïve animals. We next examined whether cholesterol supplementation could alleviate established mechanical hypersensitivity in rats with SNI, a well-characterized model of neuropathic pain (Decosterd and Woolf, 2000; Moutal et al., 2017). Intraplantar injection of WSC partially reversed mechanical hypersensitivity in the injured paw while having no effect on the noninjured contralateral paw (Fig. 1, G and H). The anti-hypersensitivity effect was sustained for 4 h. Together, these findings demonstrate that cholesterol homeostasis plays a critical role in regulating nociceptive thresholds under both physiological and neuropathic pain conditions. Moreover, considering the significant expression of HCN channels in sensory nerve fibers (Doan et al., 2004; Acosta et al., 2012), these findings provide *in vivo* evidence supporting the therapeutic potential of cholesterol enrichment and are consistent with our observations in cultured nociceptor DRG neurons (Handlin et al., 2024). Cholesterol depletion enhances pain sensitivity, whereas cholesterol supplementation reverses mechanical hypersensitivity, supporting a model in which membrane cholesterol acts as a potent modulator of pain signaling.

### Temporally resolved steps of membrane remodeling during cholesterol enrichment

To define how cholesterol modulates membrane functions that could contribute to pain sensation in neurons, we first examined changes in PM properties following cholesterol enrichment. Because cholesterol is an inherent component of OMDs, distinguishing its specific effects from those of OMDs on membrane proteins remains challenging. Altering one without affecting the other was not feasible. To quantify the OMD size, we used FRET pairs based on CTxB, specifically AF-488 CTxB conjugates as FRET donors and Alexa Fluor 647 (AF-647) CTxB conjugates as FRET acceptors (Fig. 2 A). As previously demonstrated, CTxB binds to the GM1 lipids in OMDs with high affinity (Matsubara et al., 2021; Sachl et al., 2015), and the FRET efficiency between CTxB-AF-488/CTxB-AF-647 can report the size of OMDs in cultured cells (Handlin et al., 2024). An increase in FRET efficiency reflects a higher probability that FRET donor- and acceptor-labeled CTxB molecules colocalize within the same OMD and undergo energy transfer in ensemble measurements. This increase in FRET primarily arises from the elevated likelihood that multiple CTxB molecules occupy a single OMD as OMD size expands. In contrast, small OMDs can accommodate at most one CTxB molecule, resulting in minimal FRET. A Poisson statistics–based simulation summarizes these defining features of the CTxB-based FRET approach (Fig. 2, B and C; and Fig. S1; and see Materials and methods).

To understand the roles of cholesterol itself versus the OMDs, in combination with the OMD probes, we used a genetically encoded cholesterol sensor eGFP-GRAM-W that labels the inner leaflet accessible cholesterols (Koh et al., 2023). The GRAM-W sensor is derived from the GRAM cholesterol-binding domain of the cholesterol transport protein GRAMD$_{1b}$, with a glycine

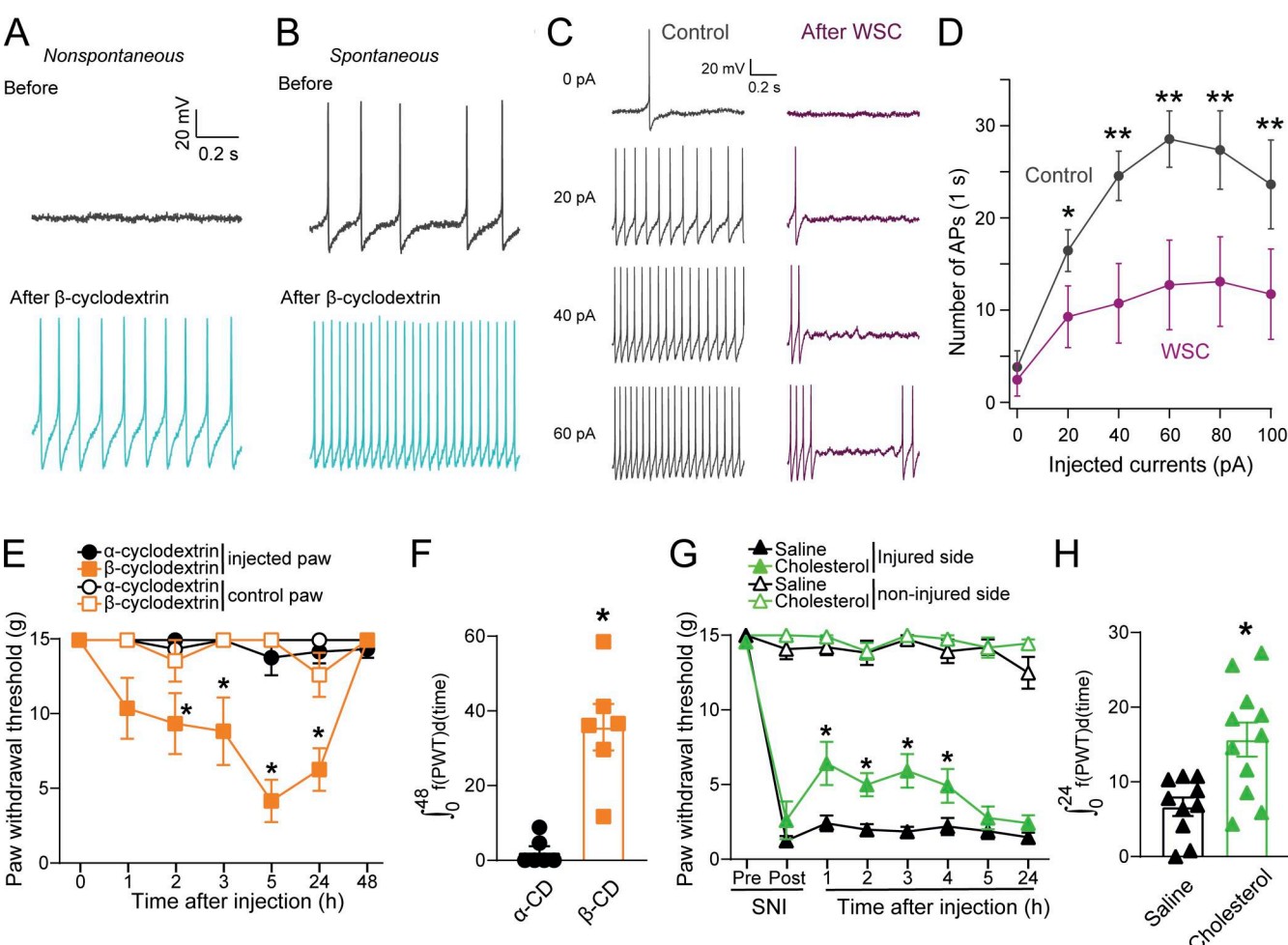

Figure 1. **Cholesterol regulates membrane excitability and mechanical hypersensitivity in naïve and neuropathic pain rats. (A–C)** Representative action potential firing of non-spontaneously (A) and spontaneously (B) firing naïve DRG neurons before and after β-CD treatment from the same-patch experiments. (C) Representative action potential firing of a naïve DRG neuron before and after the WSC treatment from the same-patch experiment. **(D)** Summary graph showing the effect of WSC on current injection-elicited action potential firing in naïve DRG neurons ($n = 11$, mean ± SEM). The P values are 0.04, 0.006, 0.005, 0.006, and 0.005 at 20, 40, 60, 80, and 100 pA current injections, respectively. **(E and F)** Naïve rats were injected intraplantarly with β-CD or its ineffective analog, α-cyclodextrin (α-CD) (20 mM each in 50 μl), and paw withdrawal threshold (PWT) was measured. In E, the graph shows the hind-paw withdrawal threshold of naïve rats injected as indicated. The threshold for the injected paw versus the uninjected paw for each group is shown. β-CD, but not α-CD, reduced the threshold in the injected paw compared with the control paw and α-CD–injected groups. The P values highlighted are 0.18, 0.06, 0.06, 0.004, and 0.004 at time points of 1, 2, 3, 5, and 24 h, comparing β-CD and α-CD conditions. In F, a bar graph with scatter plots shows the integrated area under the curves corresponding to the marked lines in panel E (from 0 to 48 h), demonstrating a significant reduction in the PWT in the β-CD–injected paw group ($n = 6$ rats per group), P = 0.002 compared with the control, Mann–Whitney test. **(G and H)** Following the SNI surgery, rats developed chronic neuropathic pain. Rats then received an intraplantar injection of WSC (~1 mg/ml, 50 μl). In G, a graph shows the PWT of rats with SNI injected as indicated. For each rat, both the injured and noninjured paws were injected, and withdrawal thresholds were measured. WSC injection significantly increased the PWT and reversed mechanical hypersensitivity on the injured side of SNI rats while having no effect on the control noninjured side. The P values highlighted are 0.02, 0.004, 2e–4, 0.03, 0.4, and 0.14 at time points of 1, 2, 3, 4, 5, and 24 h, comparing saline and cholesterol conditions. In H, a bar graph with scatter plots shows the integrated area of the data (lines with markers) in panel G (from 0 to 24 h), demonstrating a significant anti-hypersensitivity effect of cholesterol compared with saline-treated SNI rats ($n = 10–11$ rats per group), P = 0.007 compared with the control, Mann–Whitney test. Data are presented as mean ± SEM. *P < 0.05, **P < 0.01.

mutated to tryptophan (G187W) (Koh et al., 2023; Ercan et al., 2021; Naito et al., 2019). This mutation enhances its localization to the PM. Compared with traditional cholesterol sensors, such as those derived from perfringolysin O (PFO-D4) or anthrolysin O (ALO-D4), which often bind to cytosolic cholesterol, GRAM-W offers improved PM targeting and specificity (Koh et al., 2023). This makes GRAM-W a more reliable tool for studying cholesterol dynamics at the PM.

Live-cell FLIM-FRET imaging using CTxB-conjugated FRET pairs was performed after applying CTxB AF-488 (FRET donor)

and AF-647 (FRET acceptor) to label PM OMDs in tsA-201 cells—a HEK293-derived cell line commonly used for electrophysiological studies of ion channels. Fluorescence lifetimes were analyzed using phasor plots (Handlin et al., 2024), which map each pixel's lifetime and allow identification of FRET-induced shifts, reflecting an increase in the OMD size (Fig. 2, D and E). FLIM-FRET efficiency was calculated along a phasor plot-based FRET trajectory, with phasor points farther from the donor-alone position indicating higher FRET (Handlin et al., 2024). We found that the FLIM-FRET increased rapidly following the

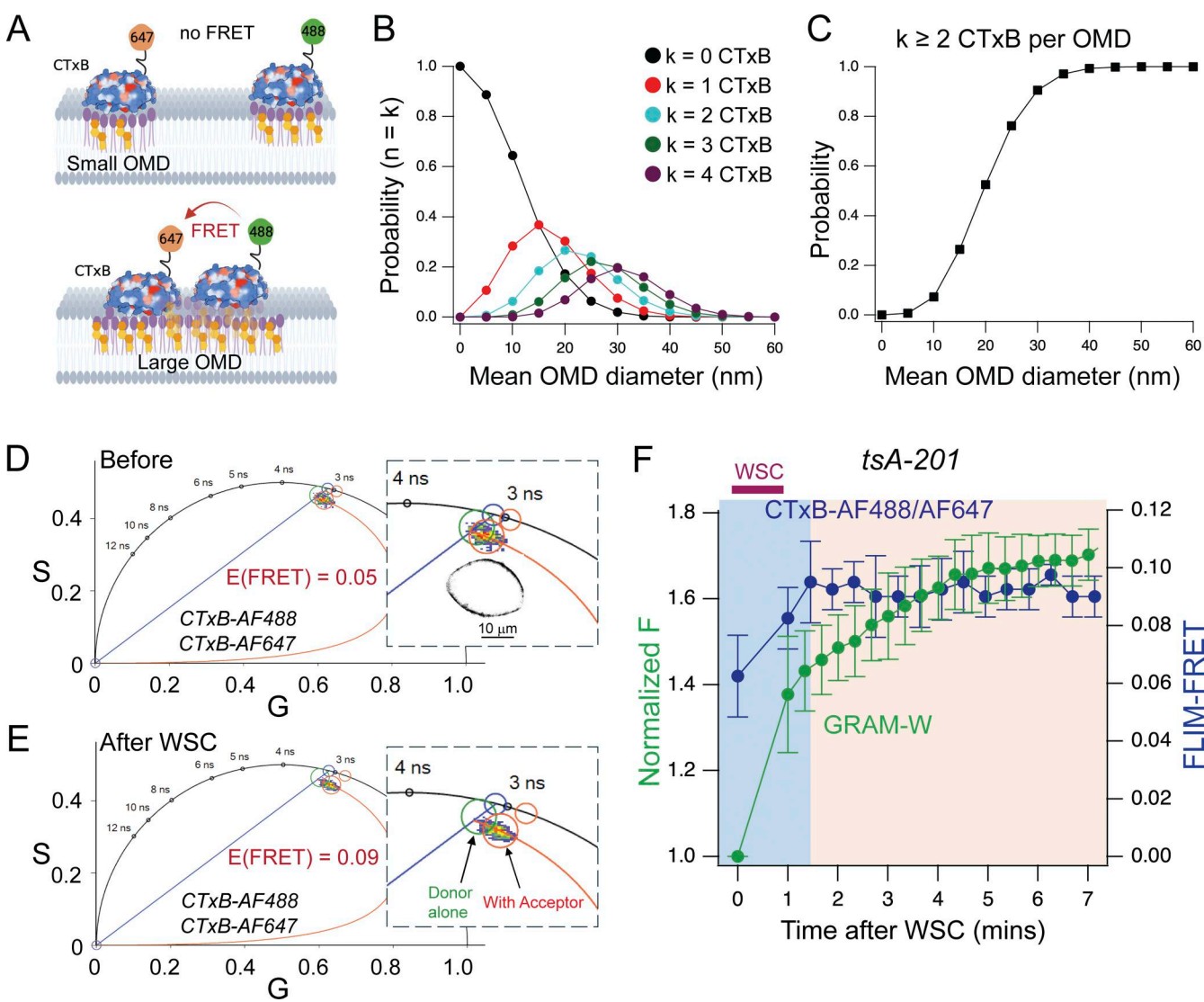

**Figure 2. Combining the CTxB-based FRET approach and a cholesterol sensor in assessing the modulation of PM properties by cholesterol enrichment. (A)** Cartoon showing the FRET between CTxB AF-488 and CTxB AF-647 under conditions of small OMDs and large OMDs. Only one fluorophore is shown for the CTxB pentamer for simplicity. **(B and C)** CTxB occupancy within OMDs was modeled using Poisson statistics with a nonlinear dependence on OMD diameter (B) to interpret CTxB-based FLIM-FRET responses to OMD expansion. The increased FRET with larger OMDs reflects a higher probability of multiple CTxB molecules occupying the same OMD (C). **(D and E)** Representative phasor plots of membrane-localized fluorescence of tsA201 cells from donor alone (CTxB AF-488 only) or with the acceptor CTxB AF-647 before (D) and after (E) the WSC treatment. The FRET efficiency was calculated using the FRET trajectory (in red). The insets show amplified views of the phasor plots. A confocal image of membrane-localized fluorescence of a tsA cell labeled with CTxB is also shown. **(F)** Time course of the change in normalized fluorescence intensity at the PM and the CTxB-based FLIM-FRET efficiency after 0.5 mg/ml WSC application in tsA201 cells. Data shown are mean + SEM; WSC was added to the culture medium at the start of time-lapse imaging, which was performed at intervals of ∼27 s for FRET ($n$ = 4) and ∼20 s for GRAM-W ($n$ = 3). Time zero refers to the steady state prior to WSC addition. The purple bar denotes the duration of WSC pipetting into the recording chamber.

addition of WSC (0.5 mg/ml cholesterol pre-complexed with mβCD) to the culture medium, reaching saturation within 2 min (Fig. 2, E and F). Although changes in GM1 density could, in principle, alter the probability of CTxB binding and thereby reduce FRET (Fig. S1), the net effect observed was an increase in FRET efficiency. These results suggest that any changes in GM1 density associated with OMD expansion are minimal and do not substantially contribute to the WSC-induced FRET changes. Instead, OMD size plays the dominant role in driving the FRET increase.

In parallel, membrane-localized fluorescence intensity of eGFP-GRAM-W also increased after WSC treatment, but with significantly slower and double-exponential kinetics (Fig. 2 F and Fig. S2). This difference suggests that OMD expansion occurs rapidly and reaches a plateau soon after cholesterol enrichment, whereas the accumulation of free cholesterol in the inner leaflet proceeds more gradually, reflecting a biphasic effect of cholesterol enrichment on PM properties and distinguishing OMD changes from those of free cholesterol. This distinction is likely due to the limited availability of other lipid species, such as

sphingolipids, for establishing OMDs, and it enables us to use this approach to functionally isolate the contributions of OMDs and free cholesterol to the PM in cultured cells.

To further demonstrate the two phases of PM modulation following cholesterol enrichment, an mCherry-tagged OlyA was used to report levels of outer leaflet cholesterol/sphingomyelin (SM) complexes (Endapally et al., 2019; Skocaj et al., 2014) (Fig. 3 A). Cholesterols sequestered by SM cannot be probed by conventional cholesterol sensors like PFO-D4 (Das et al., 2014). In addition, we employed a previously established FRET-based OMD probe to report OMD size, using a peptide (Lck 10 or L10)-based FRET pair (Sachl et al., 2015; Myeong et al., 2021; Handlin and Dai, 2023; Dai, 2022) and the CTxB-conjugated probes as described above (Handlin et al., 2024). Applying all four types of probes individually and overlaying their time courses following the same WSC treatment in tsA-201 cells consolidated the two-step effect of cholesterol enrichment on PM properties (Fig. 3 B). Similar to Fig. 2 F, CTxB- and L10-based FRET changes saturated within the first 5 min, whereas the fluorescence intensity of OlyA and GRAM-W continued to increase. Here, imaging was performed at 5-min intervals to focus on distinguishing changes in OMD-associated versus free cholesterol, rather than tracking the precise kinetics of the FRET change caused by in vitro membrane enrichment with a high cholesterol concentration. This approach particularly minimized photobleaching, which could otherwise obscure intensity-based fluorescence measurements. Consistently, we observed an initial increase in FRET between CTxB-AF-488/CTxB-AF-647 and a decrease in FRET between L10-CFP/L10-YFP. For the AF-488/AF-647 pair, the FRET efficiency increased from 9 to 11.8% (P = 0.02), whereas for the L10 FRET pair, the FRET efficiency decreased from 8.3 to 4.5% (P = 0.039) (Fig. 3 B). As previously demonstrated, these opposing FRET changes both indicate OMD expansion (Handlin et al., 2024) (Fig. 3 B). The FRET changes plateaued within 5 min, as we have shown using the higher frequency imaging in Fig. 2 F. In contrast, GRAM-W and OlyA fluorescence at the PM continued to increase beyond the first 2–5 min, extending throughout the 15–20-min timeframe (Koh et al., 2023). These results suggest that the initial phase of cholesterol enrichment involves OMD expansion along with increases in inner leaflet cholesterol and outer leaflet cholesterol/SM complexes. The second phase primarily reflects a continued accumulation of cholesterol, either in free form or bound to SM, in tsA-201 cells. Since the second phase does not involve OMD expansion, any functional effects likely arise from the free cholesterols. The limiting factor for the OMD expansion could be the amount of SM available. Collectively, this methodology provides a potential strategy to temporally distinguish the effects of OMD expansion from cholesterol itself on ion channel gating in DRG neurons.

We observed distinct effects of WSC supplementation on OMD size and free cholesterol levels in small-diameter nociceptor DRG neurons, depending on their physiological state. Specifically, neurons from naïve rats showed different patterns compared with neurons from rats subjected to the SNI model of neuropathic pain (Decosterd and Woolf, 2000), suggesting that nerve injury alters how cholesterol modulates membrane organization in these hyperexcitable sensory neurons. As we previously demonstrated, nociceptor DRG neurons in the SNI model exhibit significantly smaller OMDs compared with the large OMDs observed in naïve nociceptor DRG neurons (Handlin et al., 2024). In both conditions, membrane-localized fluorescence intensity of the GRAM-W sensor showed a sustained increase over the 15-min WSC treatment (Fig. 3, C and D). However, the CTxB-based FRET reporter of OMD size exhibited minimal changes in FLIM-FRET efficiency for naïve DRG neurons, whereas SNI DRG neurons showed a substantial increase within 5 min (Fig. 2, C and D; and Fig. S3). The initial FLIM-FRET efficiency measured using CTxB-based probes was significantly lower in SNI DRG neurons (6.3%) than in naïve neurons (15.3%) and slightly lower than in tsA-201 cells (9%). Notably, following WSC supplementation, the OMD changes detected by CTxB-based FRET in SNI DRG neurons closely mirrored those observed in tsA-201 cells, reaching an increase to 12.8%.

### Homo-FRET–based fluorescence anisotropy measurement of membrane cholesterol levels

We performed homo-FRET–based live-cell anisotropy for imaging membrane-localized GRAM-W fluorescence (Fig. 4) (Bader et al., 2011). Fluorescence anisotropy measures the polarization of emitted light by comparing perpendicular and parallel emission components, enabling the quantification of homo-FRET by detecting depolarization resulting from energy transfer between nearby fluorophores (Fig. 4, A and B). A decrease in anisotropy reflects an increase in homo-FRET efficiency, indicating closer proximity or clustering between the probes (Bader et al., 2011). Homo-FRET–based anisotropy is sensitive to the organization and density of fluorophores while being less influenced by factors such as photobleaching of fluorophores (Joshi et al., 2024). This sensitivity enables a more accurate quantification of inner leaflet cholesterol density, which would be challenging to resolve using intensity-based fluorescence alone. The lifetime of GRAM-W remained unchanged during WSC supplementation, indicating that the decrease in anisotropy was not caused by hetero-FRET or collisional quenching but is consistent with a homo-FRET mechanism (Fig. S4). Following WSC supplementation, we observed a continued decrease in GRAM-W fluorescence anisotropy, accompanied by an increase in homo-FRET efficiency (Fig. 4, C–E). Notably, the initial homo-FRET efficiency was highest in naïve DRG neurons but lower in tsA-201 cells and DRG neurons of the SNI model. These results suggest that free, accessible cholesterol in the inner leaflet is also reduced after nerve injury, adding to the previously reported decrease in OMD size observed in neuropathic pain (Handlin et al., 2024).

We also performed anisotropy experiments using OlyA-mCherry and observed distinct differences in homo-FRET responses to WSC treatment across tsA-201 cells, naïve DRG neurons, and SNI DRG neurons (Fig. 4, F–H). In tsA cells, anisotropy and homo-FRET revealed small but significant changes, aligning with the modest increase in outer leaflet cholesterol/SM complexes as concluded from the intensity-based measurements. In naïve DRG neurons, anisotropy and homo-FRET values

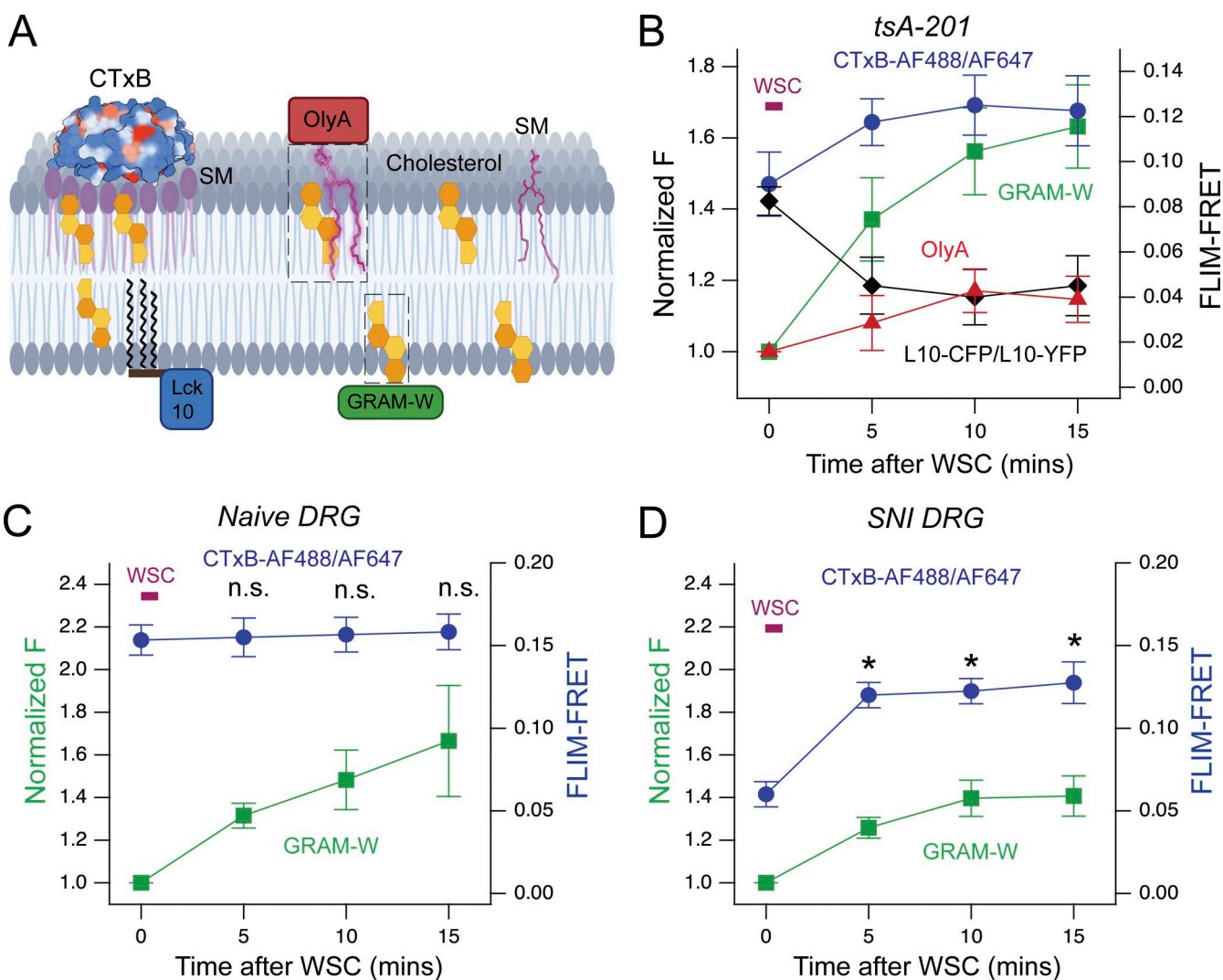

Figure 3. **Distinct modes of PM modulation by cholesterol enrichment. (A)** Strategies implemented to report the PM properties after WSC-mediated cholesterol supplementation. An mCherry-labeled OlyA was used to recognize outer leaflet cholesterol/SM complexes, whereas GRAM-W-eGFP was used to label free cholesterol levels in the inner leaflet of the PM. The OMD size was probed using FRET-based reporters, either CTxB-conjugated or peptide (L10)-based FRET pairs. **(B)** Time course of the change in normalized fluorescence intensity and the FLIM-FRET efficiency at the PM after 0.5 mg/ml WSC application in tsA201 cells, using four different probes as illustrated in panel A. Data were collected every 5 min to minimize photobleaching and shown as mean ± SEM, n = 4–10 cells. **(C and D)** Summary time course of the change in normalized GRAM-W fluorescence intensity at the PM and the FLIM efficiency of CTxB-AF488/AF647 FRET after 0.5 mg/ml WSC application for naïve rat DRG neurons (C) and for nociceptor rat DRG neurons of the SNI model (D). Data shown are mean ± SEM, n = 4–6 cells. The CTxB-based FRET level showed a significant increase after WSC for SNI DRG neurons (*P = 0.02, 0.02, and 0.03 at 5, 10, and 15 min time points using a two-sided paired $t$ test) but no change for naïve neurons (n.s., no statistical significance).

remained unchanged after WSC treatment, in contrast to the noticeable decrease in anisotropy and corresponding increase in homo-FRET observed in SNI DRG neurons. Notably, baseline homo-FRET values were highest in naïve DRG neurons and reduced in SNI DRG neurons, consistent with a decrease in OMD size following nerve injury. Given that OlyA binds to cholesterol/SM complexes enriched in OMDs, the high baseline homo-FRET and lack of response to cholesterol supplementation in naïve DRG neurons suggest that these cells already contain high or saturated levels of cholesterol/SM complexes that cannot be further increased by adding free cholesterol. This again implies that the availability of SM, rather than cholesterol, may be the limiting factor for the OMD expansion in naïve nociceptor DRG neurons.

## Inhibitory effects of cholesterol on HCN channel gating in nociceptor DRG neurons

Previously, we modulated cholesterol levels in DRG neurons for electrophysiological measurements using mβCD for extraction or WSC for supplementation, achieved by preloaded mβCD–cholesterol complexes (Handlin et al., 2024; Mahammad and Parmryd, 2015; Christian et al., 1997). These treatments were brief (30 s to 1 min) and applied prior to whole-cell patch-clamp recordings on separate control and treated cells. Here, we applied these reagents directly to the same patch during whole-cell recordings and compared hyperpolarization-activated HCN currents in nociceptor DRG neurons before and after a more prolonged treatment (Fig. 5). We found that 0.5 mg/ml WSC

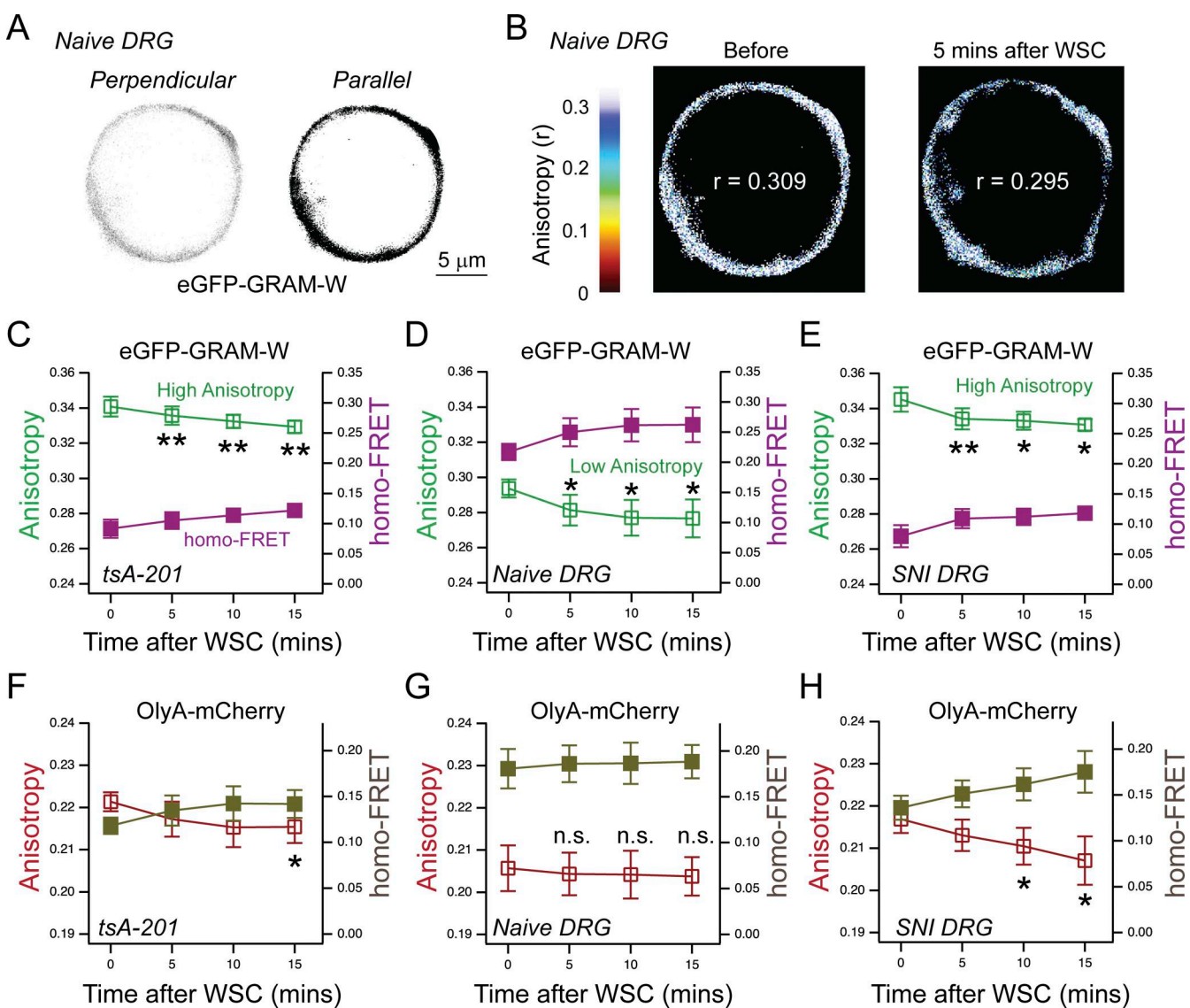

Figure 4. **Fluorescence anisotropy reports homo-FRET between cholesterol sensors. (A)** Representative confocal images showing membrane-localized eGFP-GRAW–W fluorescence emission in a naïve DRG neuron, acquired in perpendicular and parallel directions. **(B)** Corresponding anisotropy-based heatmaps derived from the images in A, shown for both pre- and post-WSC conditions, illustrating a reduction in anisotropy after WSC treatment. **(C–E)** Summary time course of the changes in GRAM-W fluorescence anisotropy and estimated homo-FRET efficiency for tsA cells (C, $n = 10$, P = 5e–4, 5e–3, and 2e–3 at 5, 10, and 15 min time points), naïve DRG neurons (D, $n = 6$, P = 0.03 for all time points), and DRG neurons of SNI model animals (E, $n = 5$, P = 3e–3, 0.01, and 0.03 at 5, 10, and 15 min time points). Since the homo-FRET data are derived from the measured anisotropy, the statistical significance is only shown for the anisotropy and remains the same for the homo-FRET. **(F–H)** Summary time course of changes in the OlyA fluorescence anisotropy and estimated homo-FRET efficiency for tsA cells (F, $n = 4$, P = 0.11, 0.08, and 0.04 at 5, 10, and 15 min time points), naïve DRG neurons (G, $n = 6$, P = 0.12, 0.2, and 0.12 at 5, 10, and 15 min time points), and DRG neurons of SNI model animals (H, $n = 4$, P = 0.08, 0.032, and 0.049 at 5, 10, and 15 min time points). Data shown are mean ± SEM, *P < 0.05, **P < 0.01 using a one-sided paired $t$ test, justified by the directional hypothesis that increased sensor crowding leads to higher homo-FRET.

supplementation significantly inhibited HCN current, reducing the amplitude of currents elicited by a maximal hyperpolarizing voltage, suggesting a decrease in channel open probability (Fig. 5, A and B), and shifting the half-maximal activation voltage leftward for the G-V relationship ($\Delta V_{1/2}$ = 8.4 mV after 10 min of WSC treatment) (Fig. 5, D and E). These effects on the G-V relationship and current amplitude persisted in the presence of additional cAMP at a supersaturating concentration of 0.5 mM in the pipette solution, with a $\Delta V_{1/2}$ of ∼ 6 mV and a ∼25% reduction in current amplitude following 10 min of WSC

treatment. The use of a supersaturating 0.5 mM concentration of cAMP (well above the ∼1 µM $EC_{50}$ for HCN channels) ensures rapid diffusion into the cytosol and a saturating effect on HCN channel activity (Kusch et al., 2010). In contrast, cholesterol extraction (5 mM mβCD) produced nearly opposite effects, increasing current amplitude (Fig. 5 C), though it had minimal impact on $V_{1/2}$, with or without the additional 0.5 mM cAMP (Handlin et al., 2024). Instead, OMD disruption by cholesterol extraction increased the slope factor ($V_s$) of the HCN current G-V relationship in naïve nociceptor neurons (Handlin et al., 2024),

whereas WSC treatment caused only a negligible change in $V_s$ in naïve neurons, both with or without 0.5 mM cAMP (Fig. 5 F). Since the $V_s$ reflects voltage sensor movement and is influenced by the OMD localization of HCN channels, it could serve as a reporter of OMD dimensions of DRG neurons (Handlin et al., 2024; Handlin and Dai, 2023). We propose that these effects arise from the naturally high cholesterol content and large OMDs in naïve DRG neurons. In this context, further increases in cholesterol levels may primarily elevate free accessible cholesterol, leading to a shift in $V_{1/2}$ and a decrease in open probability without affecting the $V_s$. Our findings lead to a model in which cholesterol modulates HCN channels through a dual mechanism: by altering OMD size and potentially interacting directly with the channels at the inner leaflet of the membrane. Moreover, this substantial modulation of HCN current amplitude (and open probability) is physiologically significant and was validated through the same patch-clamp experiments. Compared with changes in the slope factor, alterations in current amplitude may have a greater impact on action potential firing and may contribute more directly to neuronal excitability.

Beyond the equilibrium parameters of HCN gating, we found that the activation kinetics of HCN channels in nociceptor DRG neurons were modulated by cholesterol in the same patch-clamp experiments. The most prominent effect was a significant acceleration of channel activation following treatment with 5 mM mβCD, which shortened the primary time constant ($\tau_1$) of channel activation from an average of 80 ± 4.3 ms to 52 ± 2.9 ms, at the saturating hyperpolarizing voltage of –140 mV (Fig. 5 G). This effect was also present at other hyperpolarizing voltages (Fig. 5 H). In contrast, cholesterol enrichment with 0.5 mg/ml WSC for 5 min slowed channel activation, increasing $\tau_1$ from an average of 72 ± 9.6 ms to 88 ± 8.7 ms at –140 mV, although this effect was less pronounced than that of mβCD (Fig. 5, I and J). The mβCD-induced acceleration of activation persisted in the presence of 0.5 mM cAMP in the pipette solution (Fig. S5). However, under the same cAMP condition, the effect of WSC treatment on $\tau_1$ was no longer significant ($\tau_1$ from an average of 59 ± 4.5 ms to 65.3 ± 13.5 ms, $n = 4$), likely because the cAMP potentiation of channel activation is predominant, and there is likely a convergent effect between cAMP and cholesterol. Together, these findings suggest that HCN channel activation is highly sensitive to cholesterol levels. In addition, our recent findings on cholesterol-mediated modulation of current amplitude and activation kinetics (Handlin et al., 2024) were further substantiated in this study, offering new insights into the mechanisms by which HCN channels influence neuronal excitability.

### Sequential modulation of HCN channel gating by cholesterol enrichment in DRG neurons

Based on our FLIM-FRET and anisotropy experiments, indicating that the effects of cholesterol enrichment can be temporally distinguished using the WSC (0.5 mg/ml) supplementation protocol, we applied this approach to examine whether cholesterol-mediated modulation of membrane excitability following HCN activation could exhibit a biphasic feature, using a protocol in which the HCN activation in small DRG neurons was

assessed from the recovery in membrane potential after a hyperpolarizing current pulse. A –90 pA current injection induced pronounced membrane hyperpolarization, activating endogenous HCN channels, which then depolarized the membrane and restored the potential over a few hundred milliseconds to a steady state (Fig. 6 A). The percentage of voltage recovery at steady state primarily reflects the extent of HCN channel opening. We found that the percentage of voltage recovery decreased after the same-patch WSC treatment, reaching a maximal effect within ~5 min with an exponential time constant of 127 s (Fig. 6 B). No decrease was observed in the control experiments without WSC treatment (Fig. 6 B). Furthermore, similar results were observed in SNI nociceptor DRG neurons (Fig. 6 C); however, compared with naïve DRG neurons, the WSC-induced reduction showed an initial delay with a slower time constant of 184 s, consistent with a sequential cholesterol-mediated modulation of PM properties, first expanding OMDs. Presumably, in SNI neurons, voltage recovery after hyperpolarization was relatively resistant to WSC treatment during the first 200 s, likely because a substantial portion of cholesterol was initially allocated to restoring OMDs. This could result in a smaller apparent increase in the free cholesterol pool (see Fig. 3, C and D) that plays a substantial role in maintaining HCN channel opening.

Next, to examine in greater detail how cholesterol supplementation modulates HCN channel gating and to test whether this modulation occurs sequentially, we performed same-patch time course experiments. We quantified the effects of WSC (0.5 mg/ml) supplementation on HCN channel gating parameters by recording native HCN currents and plotting the G-V relationships from both naïve DRG neurons and DRG neurons of the SNI model (Fig. 6, D–I). These HCN function measurements were performed every 5 min after WSC treatment, as each voltage protocol series requires 2–3 min to complete. In naïve DRG neurons, the slope factor $V_s$ remained unchanged (Fig. 6 D). In contrast, the $V_{1/2}$ ($\Delta V_{1/2}$ = ~10 mV), current amplitude, and the primary time-constant $\tau_1$ of activation exhibited significant alterations throughout the ~15 min of WSC treatment, with the most pronounced effects observed within the first 5 min (Fig. 6, D–I). Control experiments without WSC treatment showed only minor changes, likely due to gradual dilution of endogenous cytosolic cAMP ($\Delta V_{1/2}$ = ~3 mV and little to no change in slope factor, current amplitude, and $\tau_1$ of activation, Fig. S6). Moreover, control experiments with WSC treatment, in the presence of supersaturating 0.5 mM added cAMP, showed a $\Delta V_{1/2}$ = ~7 mV and no change in the slope factor, after WSC treatment. Results from these control experiments indicate that the observed effects were specifically mediated by cholesterol enrichment.

DRG neurons from the SNI model displayed changes in all gating parameters over the course of WSC exposure (Fig. 6, G–I). The key difference compared with naïve neurons was the significant change in the $V_s$, which occurred within the first 5 min (Fig. 6 G). Additionally, the change in the $V_{1/2}$ for SNI neurons ($\Delta V_{1/2}$ = ~6 mV) was less pronounced compared with that in naïve neurons (Fig. 6, D and G; and Fig. S6), which partially explains why the WSC-induced reduction in the recovery of membrane potential in Fig 6 C was slower in SNI neurons. These

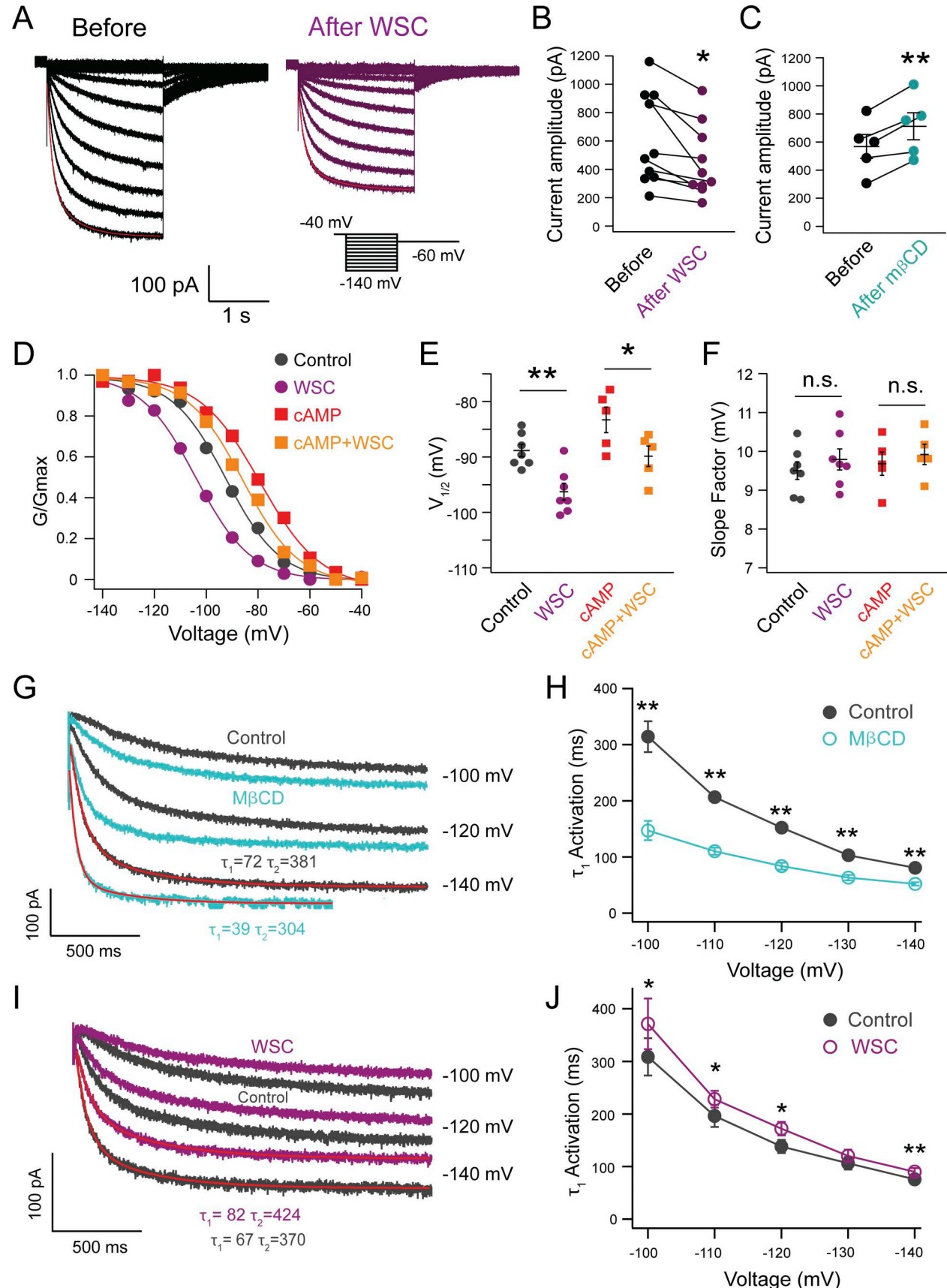

Figure 5. **Effects of manipulating cholesterol on the equilibrium and kinetic properties of HCN channel gating in nociceptor DRG neurons.** **(A)** Application of WSC decreased the current amplitude of HCN currents in the same-patch experiment. Double-exponential fittings (in red) of the current

traces at –140 mV are highlighted. **(B)** Amplitude of HCN currents elicited by a saturating hyperpolarizing voltage was used for the graph comparing the before and after treatments. Data shown are mean + SEM, $n$ = 10 patches, *P = 0.01, two-sided paired $t$ test. **(C)** Effects of the acute mβCD treatment on current amplitude of HCN currents, recorded from the same patch, showing opposite directional changes as in panel B. Data shown are mean ± SEM, $n$ = 5 patches, **P = 0.005, two-sided paired $t$ test. **(D)** Representative G-V relationship of HCN channel activation recorded from same-patch experiments before and after 5 min of 0.5 mg/ml WSC application, in the presence or absence of 0.5 mM additional cAMP. **(E and F)** Summary of the parameters derived from Boltzmann fitting of the G-V relationship of HCN channel activation before and after the WSC treatment: $V_{1/2}$ in panel E (P = 2e–5 without cAMP, P = 0.01 with cAMP) and slope factors ($V_s$) in panel F; n.s., no statistical significance. $n$ = 7 cells without added cAMP and $n$ = 5 with added 0.5 mM cAMP, two-sided paired $t$ test. **(G and I)** HCN channel currents elicited by different hyperpolarizing voltages were shown with double-exponential fits to highlight the acceleration of channel activation by mβCD (G) and the slowdown of channel activation by WSC (I). **(H and J)** Summary graph illustrating the effects of mβCD (H, $n$ = 5, P = 3e–3, 6e–5, 4e–5, 1e–3, and 1e–4 for 100 mV, –110 mV, –120 mV, –130 mV, and –140 mV) and WSC (J, $n$ = 8, P = 0.02, 0.02, 0.016, 0.18, and 1e–3 for 100 mV, –110 mV, –120 mV, –130 mV, and –140 mV) on the $\tau_1$ of channel activation at different voltages. Data shown are mean ± SEM, *P < 0.05, **P < 0.01, two-sided paired $t$ test.

findings suggest that the reduction in the slope factor primarily reflects OMD expansion, whereas the changes in $V_{1/2}$, current amplitude, and $\tau_1$ for activation likely result from both OMD expansion and an increased accumulation of accessible cholesterols in the PM. Similarly, all gating parameters changed more markedly within the first 5 min than between 5 and 15 min (Fig. 6, D–I). Cholesterols could exert direct binding effects on HCN channels in DRG neurons, as suggested by a putative cholesterol binding site of HCN3 channels (Yu et al., 2024), which is conserved across all HCN isoforms.

We previously demonstrated that the voltage sensor S4 helix of the HCN channel is sensitive to changes in OMD size in DRG neurons (Handlin and Dai, 2023; Handlin et al., 2024). Notably, the unusually long S4 segment in HCN channels (Lee and MacKinnon, 2017; Lee and MacKinnon, 2019; Handlin and Dai, 2023) may underlie their unique susceptibility to dual modulation by freely accessible cholesterol and OMDs, a feature not commonly observed in other voltage-gated ion channels. To assess whether this mechanism extends to other key ion channels involved in DRG neuron excitability, we recorded sodium currents from small nociceptive DRG neurons following cholesterol depletion (Fig. S7). Sodium channels in DRG neurons play important roles in controlling action potential firing and pain sensation (Alsaloum et al., 2025; Catterall, 2014). Similar to previous findings (Albani et al., 2024; Amsalem et al., 2018), mβCD treatment caused a slight hyperpolarizing shift in the activation curve ($\Delta V_{1/2}$ = ~5 mV), without affecting the slope factor of activation or steady-state parameters of the fast inactivation. In contrast, as we emphasized earlier, cholesterol depletion increases the slope factor of the G-V relationship for HCN channels in nociceptor DRG neurons, suggesting a decrease in the effective gating charge movement associated with channel activation (Handlin et al., 2024). Sodium channels may not preferentially localize to OMDs, or their shorter S4 segments may lack sensitivity to changes in membrane domain compartmentalization (Wisedchaisri et al., 2019). These findings suggest that the dual modulatory effect of cholesterol on HCN channels may represent a channel-specific mechanism not shared by voltage-gated sodium channels.

## Discussion

Our approach investigates the effects of cholesterol on ion channels and membrane properties. By combining FLIM-FRET–based imaging of OMDs, cholesterol sensors, and homo-FRET–based anisotropy imaging, we have developed a strategy to temporally distinguish the contributions of OMDs from those of free, accessible cholesterol in the PM. This approach can be broadly applied to study other membrane proteins regulated by lipid microenvironments, such as voltage-gated potassium channels, G-protein–coupled receptors, and mechanosensitive ion channels. Many of these proteins localize to or interact with OMDs, yet their regulation by cholesterol remains poorly understood. By dissecting dynamic changes in compartmentalized nanodomains and cholesterol distribution, our methodology enables future studies to explore how cholesterol modulates ion channels and impinges on neuronal excitability.

In neuropathic pain, upregulation of HCN channels and shifts in their voltage dependence contribute to spontaneous firing and heightened pain sensitivity (Emery et al., 2011; Tsantoulas et al., 2017; Young et al., 2014). By identifying a dual mechanism by which cholesterol regulates HCN channels—via both OMD expansion and increased inner leaflet cholesterol—we provide a mechanism for understanding how membrane lipid composition alters excitability in disease states. HCN channel hyperactivity is a well-established facilitator of neuronal firing; its inhibition—such as through cholesterol-mediated modulation—can effectively dampen the pathological hyperexcitability. However, the role of HCN is highly context dependent: when activated near resting potential, opening HCN channels can often shunt excitability by increasing membrane conductance, effectively suppressing depolarizing currents (Poolos et al., 2002; Hu and Bean, 2018; Ko et al., 2016; Tsantoulas et al., 2017; Young et al., 2014; Vasylyev et al., 2023; George et al., 2009). This suggests that cholesterol may not uniformly mitigate excitability in all neurons; its effect likely depends on the specific cellular context, including the resting membrane potential, HCN isoform expression, and concurrent activation of other voltage-gated channels. This context dependence determines whether the dominant effect of HCN opening is excitatory or inhibitory for membrane excitability. Specifically, our results demonstrate that a decrease in free cholesterol levels at the membrane of nociceptor DRG neurons following nerve injury may be functionally linked to pain behaviors. We demonstrate that cholesterol depletion sensitizes naïve animals to mechanical stimuli, whereas cholesterol supplementation attenuates mechanical hypersensitivity in a neuropathic pain model, consistent with an inhibitory role of cholesterol. These pain-related phenotypes are accompanied by reduced homo-FRET between cholesterol sensors and distinct alterations in HCN channel gating, including

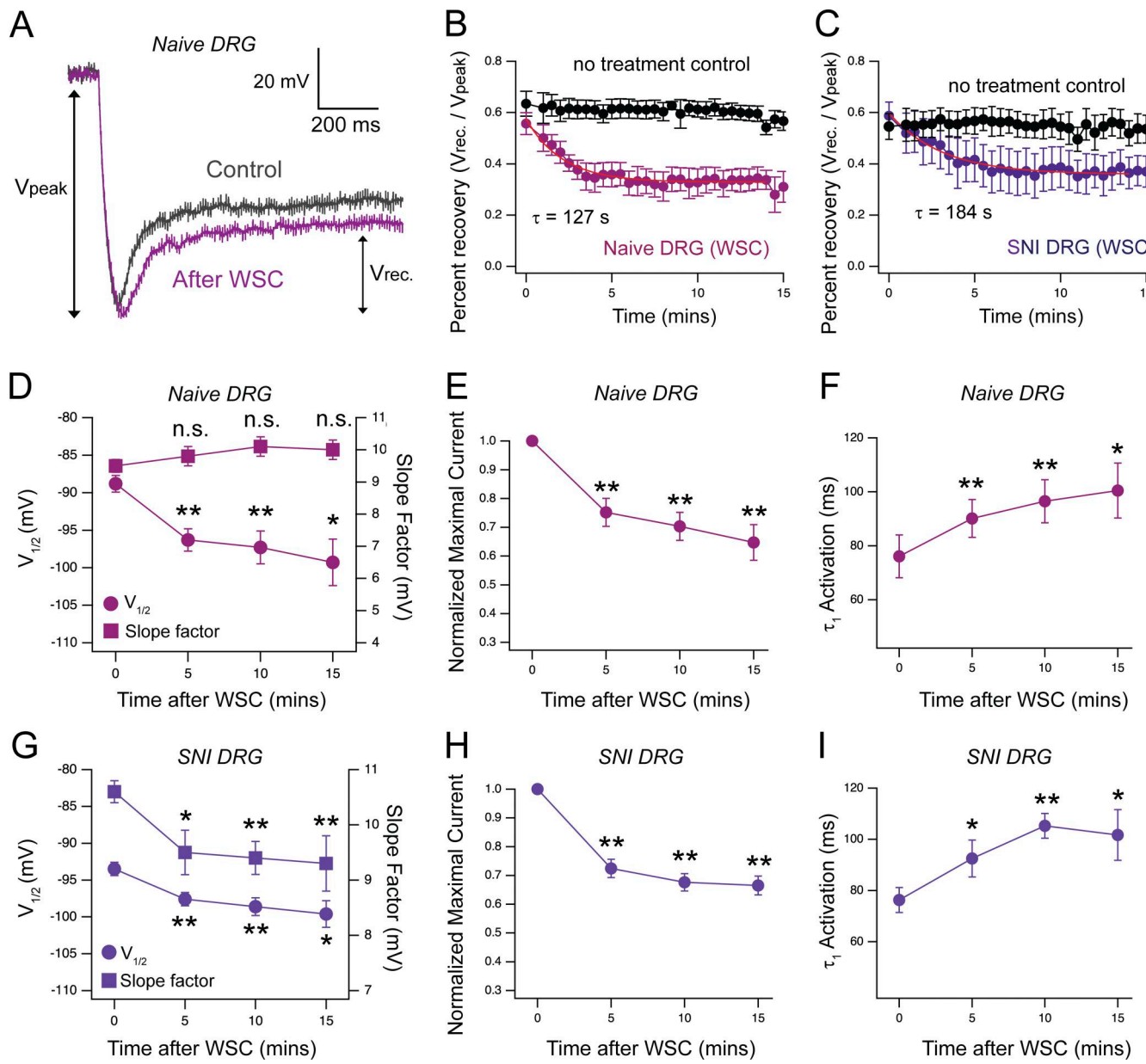

Figure 6. **Effects of cholesterol enrichment on HCN channel function in nociceptor DRG neurons from naïve and SNI animals. (A)** Representative protocol and membrane potential recordings using hyperpolarizing current injection (−90 pA) to activate HCN channels, leading to membrane depolarization. The ratio of the recovered potential ($V_{rec.}$) relative to the total hyperpolarization ($V_{peak}$) reflects the percent of recovery in membrane potential mediated by HCN activation. **(B)** Time course of changes in the percent recovery ($V_{rec.}/V_{peak}$) after WSC treatment for naïve nociceptor DRG neurons ($n$ = 5 patches), compared with untreated controls ($n$ = 6 patches), mean ± SEM; red trace shows a single-exponential fit. **(C)** Time course of the change in the percent recovery for ipsilateral SNI DRG neurons ($n$ = 4 patches with WSC treatment, mean ± SEM) using the same protocol as in panels A and B, compared with the no-treatment controls using ipsilateral neurons ($n$ = 4 patches). **(D–F)** Time course of the change in the $V_{1/2}$ and slope factor ($V_s$) derived from the G-V relationship (D, P = 2e−5, 4e−4, and 0.02 at 5, 10, and 15 min time points for $V_{1/2}$), the current amplitude (E, P = 6e−4, 2e−4, and 7e−4 at 5, 10, and 15 min time points), and $\tau_1$ of channel activation (F, P = 6e−4, 1e−3, and 0.01 at 5, 10, and 15 min time points) after 0.5 mg/ml WSC application for naïve small DRG neurons; n.s., no statistical significance for the slope factor over time after WSC treatments, $n$ = 7–10 patches. **(G–I)** Summary time course of the change in the $V_{1/2}$ and slope factor ($V_s$) (G, P = 1e−4, 9e−4, and 4e−3 at 5, 10, and 15 min time points for $V_{1/2}$, P = 0.004, 9e−5, and 0.014 at 5, 10, and 15 min time points for the slope), the current amplitude (H, P = 1e−4, 4e−5, and 1e−4 at 5, 10, and 15 min time points), and $\tau_1$ of channel activation (I, P = 0.03, 0.0017, and 0.026 at 5, 10, and 15 min time points) after 0.5 mg/ml WSC application for SNI small DRG neurons. Compared with naïve neurons (D), the slope factor in panel G showed significant changes in SNI neurons following WSC treatment. Data shown are mean ± SEM, $n$ = 4–7 patches, *P < 0.05, **P < 0.01, two-sided paired $t$ test.

shifts in voltage sensitivity and activation kinetics. The ability to temporally separate the effects of OMD expansion from those of free cholesterol accumulation using biophysical and imaging tools allowed us to link specific gating changes to different stages of membrane remodeling. Notably, changes in slope factor ($V_s$) were more closely associated with OMD expansion, while shifts in half-maximal activation voltage ($V_{1/2}$) and time constants were more sensitive to increased cholesterol content in the inner leaflet.

This dual mechanism of HCN channel modulation likely arises from their distinctive structural features. The long S4 voltage-sensing helix in HCN channels, which spans a considerable portion of the membrane electric field, is particularly responsive to changes in membrane order and lipid packing (Lee and MacKinnon, 2017; Lee and MacKinnon, 2019; Handlin and Dai, 2023). Furthermore, the presence of putative cholesterol-binding motifs in HCN channel subtypes raises the possibility of direct cholesterol–channel interactions that modulate gating. In addition, cholesterols could regulate other membrane lipids and indirectly modulate channel activities (Kwarteng et al., 2025; Delgado-Ramirez et al., 2024; Taglieri et al., 2012; Pike and Miller, 1998), which remains an area of future investigation. These features make HCN channels highly susceptible to lipid-mediated modulation, offering a mechanistic insight into their dynamic behavior in sensory neurons under inflammation or injury. From a translational perspective, these findings raise the possibility that manipulating membrane cholesterol—either pharmacologically or genetically—could serve as a strategy for tuning HCN channel function and alleviating chronic pain. While current therapies targeting HCN channels have largely focused on pharmacological pore blockers, our work suggests that modulating the HCN voltage sensor/OMD interactions or the lipid-bilayer nano-environment could achieve similar outcomes.

In summary, this study provides insights into the biophysical mechanisms by which free cholesterol and OMD-associated cholesterol pools cooperatively regulate ion channel activity, with relevance for HCN channel function in nociceptors. By integrating fluorescence imaging, electrophysiology, and *in vivo* behavior, we present a view of how sequential cholesterol-driven membrane lipid remodeling contributes to neuronal hyperexcitability and pain. This integrative paradigm offers new insights for understanding and treating neuropathic pain by targeting the membrane environment.

## Data availability

All source data in Figs. 1, 2, 3, 4, 5, and 6 are provided as a Source Data file in figshare.com as https://doi.org/10.6084/m9.figshare.29983183, and additional data are in the online supplemental material.

## Acknowledgments

Crina M. Nimigean served as editor.

Cartoons in figures were created using BioRender.

This research is supported by the National Institute of General Medical Sciences R35GM154778 grant (to G. Dai) and by the Doisy Fund (to G. Dai and L.J. Handlin) of the Edward A. Doisy Department of Biochemistry and Molecular Biology at Saint Louis University School of Medicine. The Moutal lab is supported by startup funds from the Department of Pharmacology and Physiology, Saint Louis University; SLU Institute for Drug and Biotherapeutic Innovation seed grant; National Institute of Neurological Disorders and Stroke R01NS119263 grant; and the Department of Defense Chronic Pain Management Research Program #CP220080P1 grant.

Author contributions: Lucas J. Handlin: data curation, formal analysis, investigation, methodology, project administration, validation, visualization, and writing—review and editing. Clémence Gieré: investigation. Nicolas L.A. Dumaire: investigation. Lyuba Salih: investigation and methodology. Aubin Moutal: formal analysis, investigation, resources, and writing—original draft, review, and editing. Gucan Dai: conceptualization, data curation, formal analysis, funding acquisition, investigation, methodology, project administration, resources, supervision, validation, visualization, and writing—original draft, review, and editing.

Disclosures: The authors declare no competing interests exist.

Submitted: 4 November 2025

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

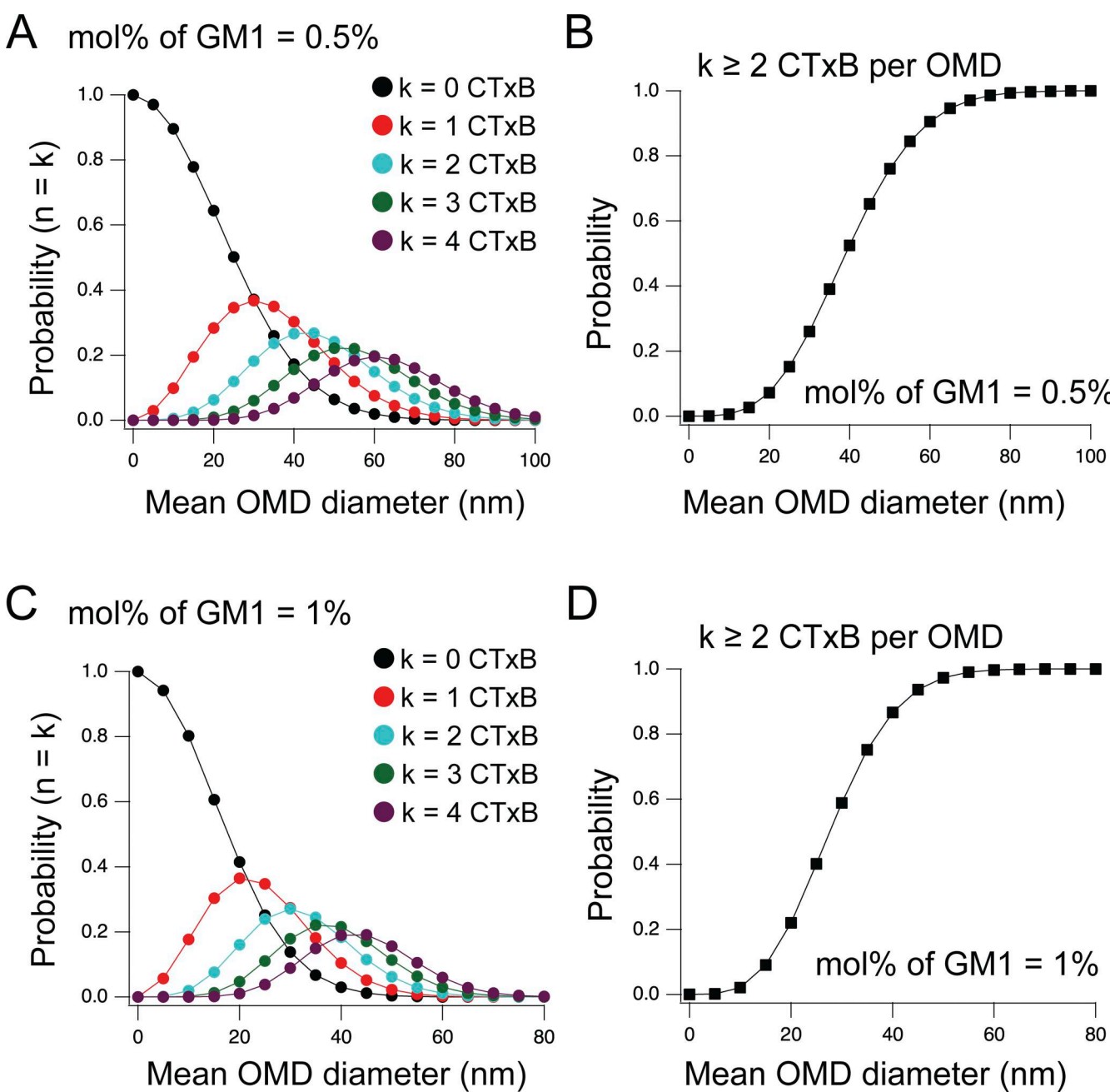

Figure S1. **Poisson statistics–based simulations illustrating the increased probability of CTxB binding as OMD size increases. (A–D)** Relative to simulations using 2 mol% GM1 (Fig. 2), simulations shown here assume lower GM1 densities within OMDs: 0.5 mol% (A and B) and 1 mol% (C and D). Within a given cell type, GM1 density within OMDs is assumed to remain constant.

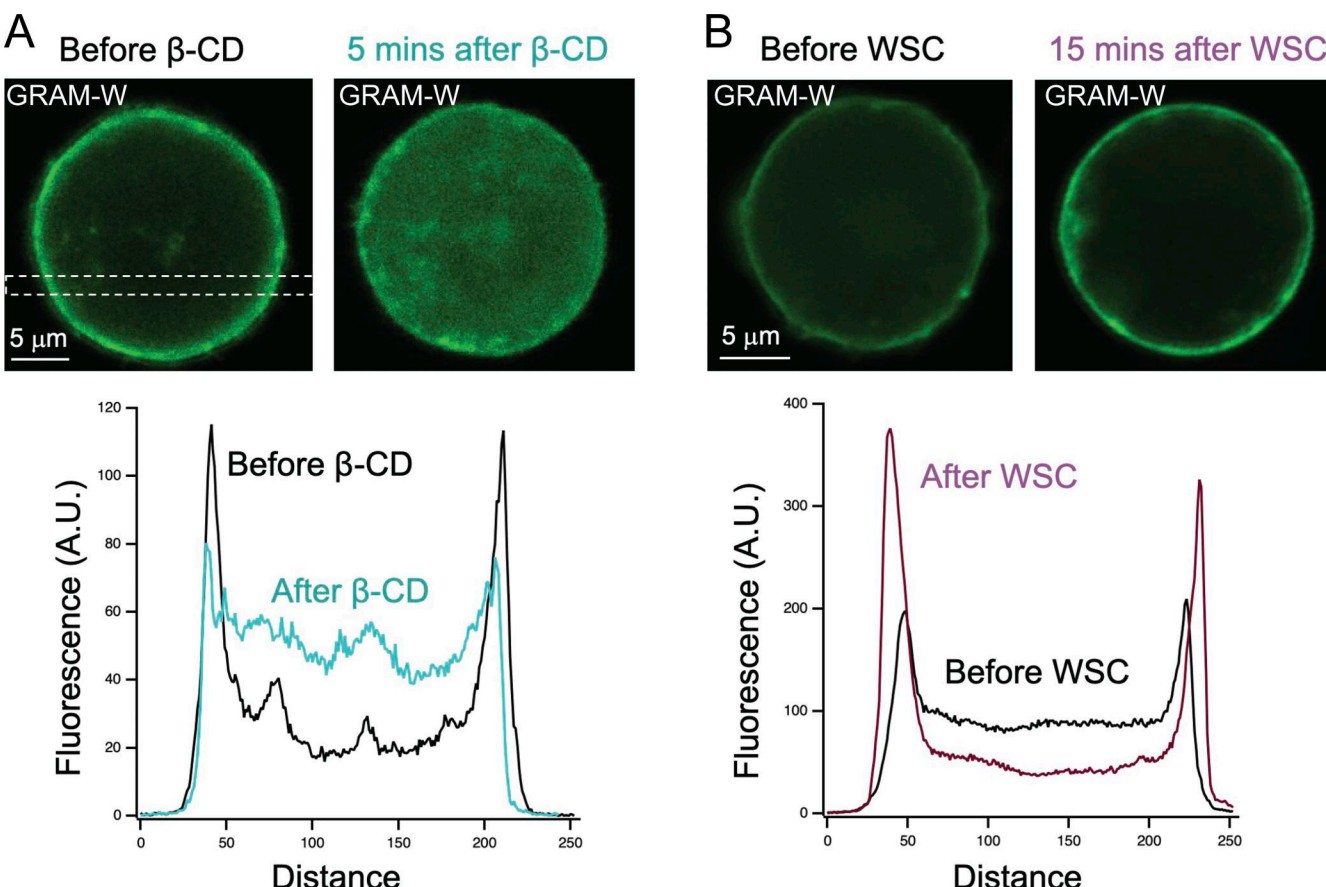

Figure S2.  **Confocal imaging showing the effects of cholesterol extraction or supplementation on membrane localization of GRAM-W-eGFP. (A)** tsA cells overexpressing GRAM-W-eGFP, before and after 5 min of treatment with 2.5 mM β-CD. Line scans of an area such as that indicated by white dashed lines are plotted to highlight the increase in cytosolic GFP fluorescence after β-CD. **(B)** tsA cells overexpressing GRAM-W-eGFP, before and after 15 min of treatment with 0.5 mg/ml WSC, highlighting the increase in membrane-localized GFP fluorescence after cholesterol supplementation.

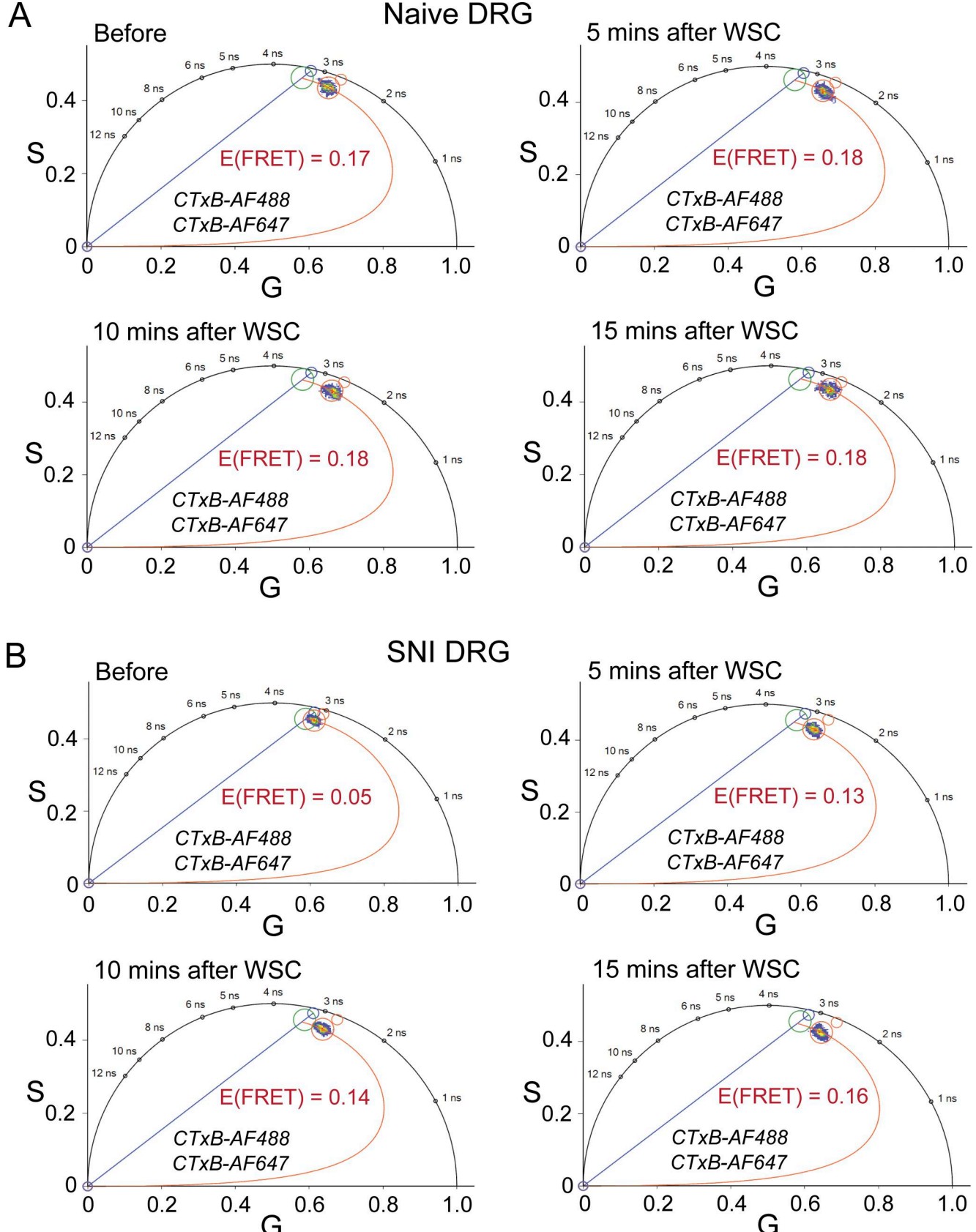

**Figure S3.** **CTxB-based FLIM-FRET responses to cholesterol enrichment in naïve and SNI nociceptor DRG neurons. (A and B)** Representative phasor plots showing membrane-localized fluorescence in naïve (A) and SNI (B) nociceptor DRG neurons labeled with CTxB AF-488 and CTxB AF-647, recorded before and at 5, 10, and 15 min after WSC treatment.

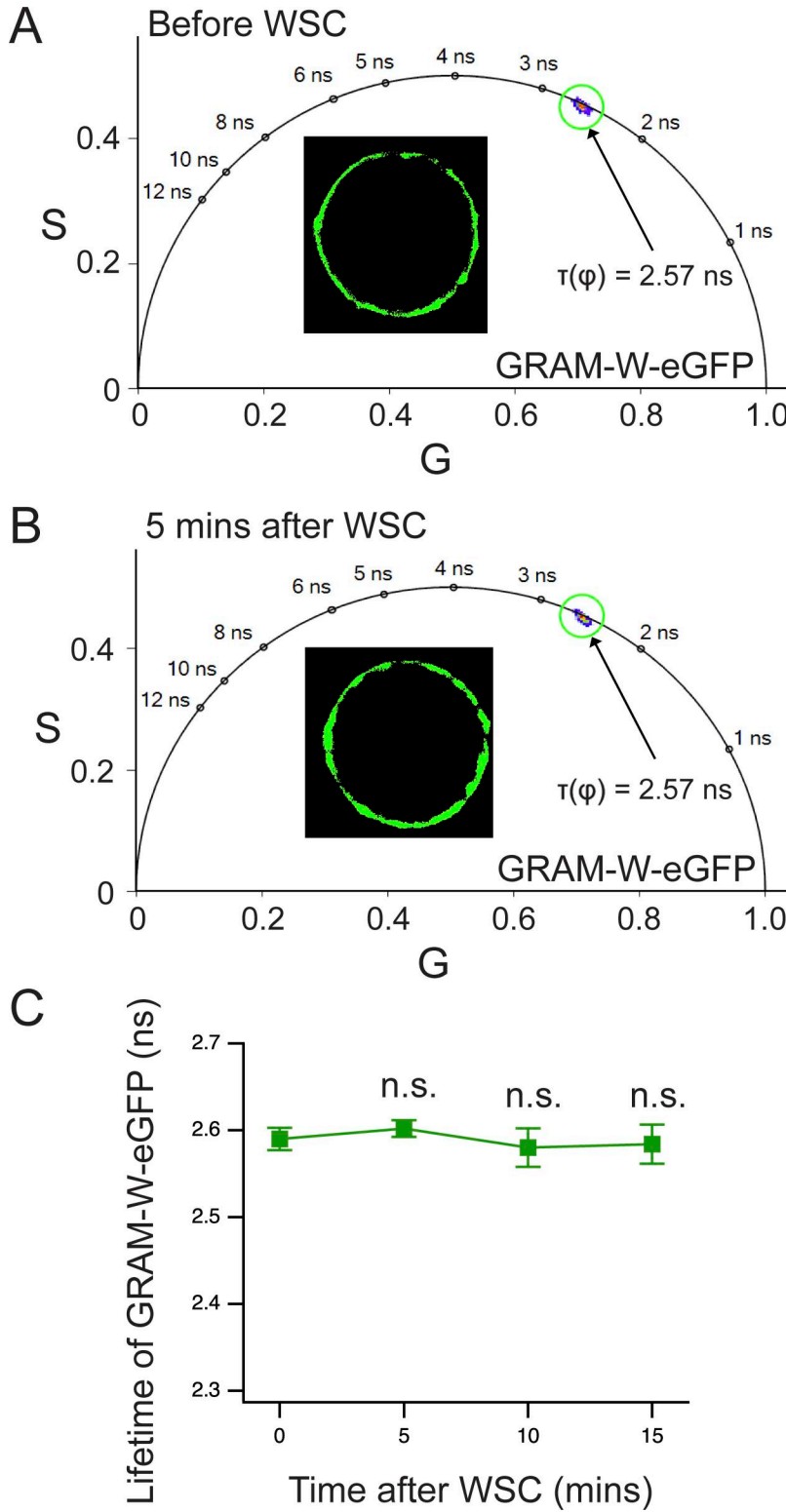

Figure S4.  **WSC treatment does not change the fluorescence lifetime of eGFP-GRAM-W. (A and B)** Phasor analysis of eGFP-GRAM-W at the PM of tsA cells before and 5 min after WSC treatment. Plots depict fluorescence distribution and the corresponding apparent phase lifetime ($\tau\varphi$). **(C)** Time course of the mean phase lifetime ($\tau\varphi$) shows no change over 15 min of WSC treatment (mean ± SEM, $n$ = 5 cells; n.s., not significant using a paired $t$ test). During this period, eGFP-GRAM-W exhibited a decrease in fluorescence anisotropy and an increase in intensity, consistent with sensor clustering and increased local density. In contrast, restriction of GFP mobility due to aggregation would be expected to increase anisotropy. Moreover, the stability of the fluorescence lifetime under these conditions rules out a change in rotational mobility as the cause of the anisotropy decrease. Instead, these data are indicative of fluorescence depolarization via an increase in homo-FRET efficiency.

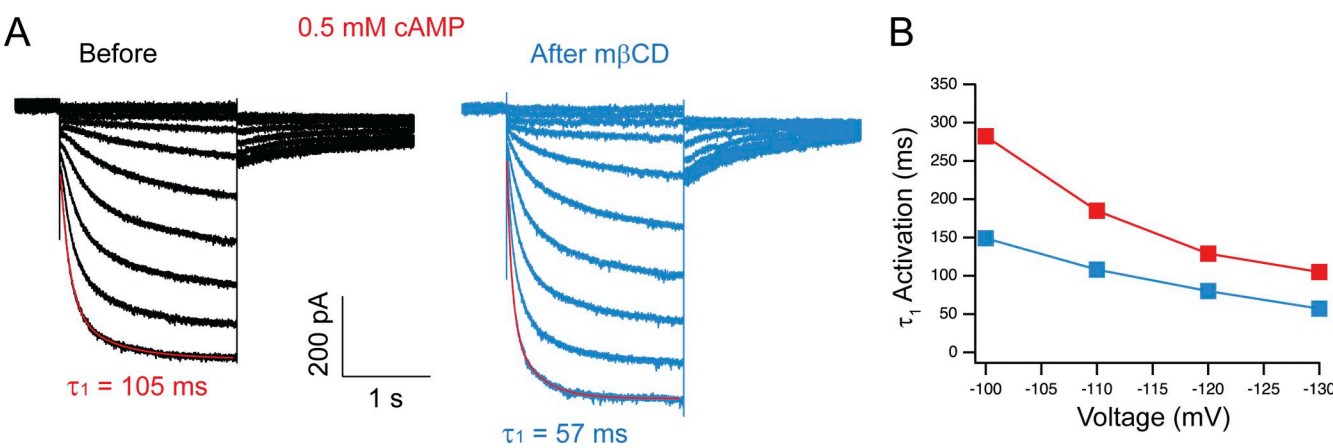

Figure S5. **Increase in current amplitude and acceleration of the HCN channel activation by cholesterol depletion in the presence of a saturating concentration of cAMP. (A)** Representative whole-cell HCN current traces recorded from naïve nociceptor DRG neurons in the presence of the added 0.5 mM cAMP in the patch-clamp pipette solution. Current amplitude increased, and the kinetics of channel activation were faster. **(B)** Measured primary time constant of channel activation at different hyperpolarizing voltages from the same patch in the panel A.

Figure S6. **Control experiments examining HCN channel gating in naïve DRG nociceptors without WSC treatment, in comparison with conditions with WSC treatment. (A–C)** Summary time course of the change in the $V_{1/2}$ and slope factor ($V_s$) (A), the current amplitude (B), and $\tau_1$ of channel activation (C) after 0.5 mg/ml WSC application for naïve DRG neurons. Data shown are mean ± SEM, $n$ = 7 patches. **P = 0.004 in panel A using two-sided paired $t$ test. **(D)** Summary of the change in $V_{1/2}$ ($\Delta V_{1/2}$) 5 min after with or without WSC treatment for naïve and SNI DRG neurons. Added 0.5 mM cAMP in the pipette solution was also included as a comparison, showing no difference from the condition without the added cAMP, suggesting the effect was mediated by WSC. Data shown are mean ± SEM, $n$ = 5–7 patches, one-way ANOVA, **P = 1e–4 (naïve: WSC versus no treatment); **P = 0.007 (naïve versus SNI, with WSC).

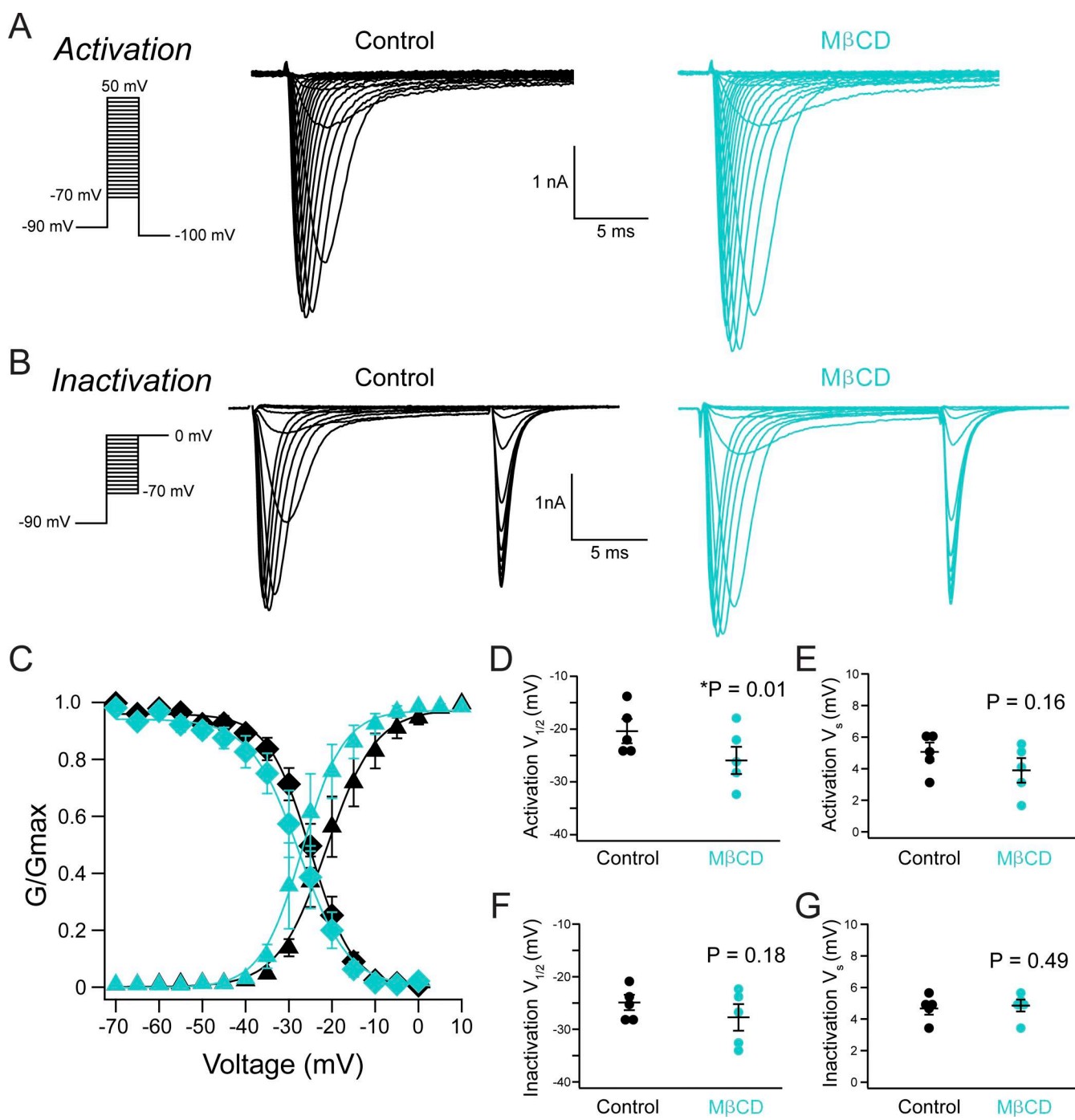

Figure S7. **Effects of cholesterol extraction on the gating of endogenous sodium currents of naïve nociceptor DRG neurons. (A and B)** Representative currents caused by sodium channel activation (A) and inactivation (B) in naïve nociceptor DRG neurons, before and after acute 2.5 mM mβCD treatment. **(C)** Averaged G-V relationships showing the activation (triangle maker) and inactivation (diamond shape) profiles of endogenous sodium channels before and after acute 2.5 mM mβCD treatment. **(D and E)** Summary data showing the $V_{1/2}$ and the slope factor ($V_s$) for sodium channel activation. **(F and G)** Summary data showing the $V_{1/2}$ and the slope factor ($V_s$) for sodium channel inactivation. Data shown are mean ± SEM, $n$ = 5 patches, *P = 0.01 using two-sided paired $t$ test.

**Provided online is Table S1. Table S1 shows effects on HCN gating parameters by cholesterol supplementation.**

