## [Peer Review File · The Journal of General Physiology]

Sequential Membrane Remodeling by Cholesterol Distinctly Modulates HCN Channels in Naïve and Neuropathic DRG Neurons

Lucas Handlin, Clémence Gieré, Nicolas Dumaire, Lyuba Salih, Aubin Moutal, and Gucan Dai

Corresponding Author(s): Gucan Dai, Saint Louis University

Review Timeline:

Submission Date:	November 4, 2025
Editorial Decision:	December 10, 2025
Revision Received:	January 21, 2026
Editorial Decision:	February 17, 2026
Revision Received:	February 18, 2026

Editor: Crina Nimigean

Transaction Report:

DOI: <https://doi.org/10.1085/jgp.202513925>

December 10, 2025

Dr. Gucan Dai
Saint Louis University
Department of Biochemistry and Molecular Biology
School of Medicine
St. Louis, MO 63104

Re: 202513925

Dear Gucan,

Thank you for submitting your manuscript, entitled "Sequential Membrane Remodeling by Cholesterol Distinctly Modulates HCN Channels in Naïve and Neuropathic DRG Neurons" to JGP. Your manuscript has now been seen by 3 reviewers, whose comments are appended below. You will see that all three reviewers are enthusiastic about the novelty and significance of your findings and the considerable promise of the techniques employed to evaluate membrane processes. However, they also raised several concerns that should be addressed prior to further consideration of the manuscript at JGP. In particular, please view the editor's summary directly below.

Editor Summary

Reviewers and editors are in agreement that the study addresses an important and experimentally challenging question—namely, how cholesterol modulates HCN1 channels in DRG neurons. They find the electrophysiological data compelling, the integration with pain models intriguing, and the manuscript generally well written with extensive datasets. However, several substantive issues must be addressed before the manuscript can be considered further.

1. FRET-FLIM Analysis of OMD Size: Reviewers expressed concern that the FRET-FLIM approach requires clearer justifications and additional controls. The cited literature provides only limited and model-dependent relationships between FRET signal and OMD radius, typically in simplified, protein-free membranes. The authors should:

- Add first a conceptual description of how OMD size estimation can be derived from FLIM-FRET using Alexa-tagged CTxB
- Provide stronger methodological justification or additional validation controls for the relation between size and FRET signal;
- Clarify how presence of membrane proteins in bilayer may influence CTxB binding and the OMD size interpretation;

2. Support for the "Sequential Cholesterol-Mediated Modulation of HCN Function" Conclusion: Reviewers and editors found it difficult to identify which data specifically support this central conclusion. The manuscript needs a clearer, more explicit linkage between the experimental findings and this proposed sequence of modulation. Relatedly, the assignment of time constants in the SNI DRG experiments (e.g., 176 s in text vs. 184 s in Fig. 4c) should be corrected and explained in this context.

3. Interpretation of Cholesterol Effects on HCN Gating: Although the general conclusions are understandable, the detailed interpretation of how cholesterol alters individual HCN gating properties is not sufficiently clear. The manuscript should provide a more focused and systematic explanation of how cholesterol addition versus removal affects activation, deactivation, and recovery kinetics. A table summarizing the effects of cholesterol on gating parameters (e.g., $P_{o,max}$, $V_{1/2}$, V_s , τ_1), indicating whether each effect is more likely due to a change in OMD size or free cholesterol would be beneficial. In addition, sometimes the effects on HCN channels with cholesterol removal were opposite of the effects observed when cholesterol was added. It remains unclear whether these manipulations exert opposing effects on cholesterol distribution across membrane pools. This point requires deeper discussion, ideally referencing the sequential membrane remodeling framework and how redistribution among cholesterol pools may explain differential gating effects.

4. Integration of Mechanistic and Behavioral Findings: Reviewers noted a disconnect between the mechanistic membrane/HCN experiments and the DRG excitability and mechanical allodynia data. To improve coherence, the authors should either 1) strengthen the mechanistic link by discussing the relative contributions of distinct cholesterol pools to nociceptive thresholds, including how these contributions may change in the SNI model; or 2) consider restructuring the manuscript so that the pain-behavior and DRG excitability data appear first, followed by the mechanistic investigation.

5. Organization and Focus: The manuscript would benefit from improved organization and a more unified narrative. The abstract would benefit from a rewrite as it could emphasize better the central findings and conclusions of the manuscript to guide the readers through the paper. We also recommend highlighting early on and more explicitly the connection between sequential membrane remodeling with cholesterol and the differential effects of adding versus removing cholesterol on HCN gating.

6. Evidence for HCN1 Association with OMDs: The evidence for HCN1 localization within OMDs should be more clearly presented. Consider including this material in the Introduction or early Results to frame the mechanistic rationale for the study.

7. Statistics: Please add exact p values to all figures.

We would be pleased to receive a suitably revised manuscript that addresses these concerns, which will be re-reviewed, most likely by some or all of the original referees. In addition, please do not hesitate to contact me (via the editorial office) if you feel

that a discussion of the reviewers' and editors' comments would be helpful.

Please submit your revised manuscript via the link below along with a point-by-point letter that details your responses to the editor's summary and reviewers' comments, as well as a copy of the text with alterations highlighted (boldfaced or underlined). If the article is eventually accepted, it would include a 'revised date' as well as submitted and accepted dates. If we do not receive the revised manuscript within one year, we will regard the article as having been withdrawn. We would be willing to receive a revision of the manuscript at a later time, but the manuscript will then be treated as a new submission, with a new manuscript number.

Please pay particular attention to recent changes to our instructions to authors in the following sections: Data presentation, Blinding and randomization and Statistical analysis, under Materials and Methods, as shown here: <https://rupress.org/jgp/pages/submission-guidelines#prepare>. Re-review will be contingent on inclusion of the required information (including for data added during revision) and demonstration of the experimental reproducibility of the results. Also, to improve the reproducibility of published content, we have partnered with SciScore. Authors are prompted in eJP to copy and paste the Materials and Methods section of their manuscript for a SciScore assessment when submitting their revised manuscript. Authors are encouraged (not required) to further revise their Materials and Methods if the SciScore is below 4. More information can be found here: <https://rupress.org/jgp/pages/submission-guidelines#sciscore>

Please note, JGP requires authors to submit Source Data used to generate figures containing gels and Western blots with all revised manuscripts (when applicable). This Source Data consists of fully uncropped and unprocessed images for each gel/blot displayed in the main and supplemental figures. If your paper includes cropped gel and/or blot images, please be sure to provide one Source Data file for each figure that contains gels and/or blots along with your revised manuscript files. File names for Source Data figures should be alphanumeric without any spaces or special characters (i.e., SourceDataF#, where F# refers to the associated main figure number or SourceDataFS# for those associated with Supplementary figures). The lanes of the gels/blots should be labeled as they are in the associated figure, the place where cropping was applied should be marked (with a box), and molecular weight/size standards should be labeled wherever possible. Source Data files will be made available to reviewers during evaluation of revised manuscripts and, if your paper is eventually published in JGP, the files will be directly linked to specific figures in the published article.

Source Data Figures should be provided as individual PDF files (one file per figure). Authors should endeavor to retain a minimum resolution of 300 dpi or pixels per inch. Please review our instructions for export from Photoshop, Illustrator, and PowerPoint here: <https://rupress.org/jgp/pages/submission-guidelines#revised>

When revising your manuscript, please be sure it is a double-spaced MS Word file and that it includes editable tables, if appropriate.

Please submit your revised manuscript via this link:
Link Not Available

Thank you for the opportunity to consider your manuscript.

Sincerely,

Crina Nimigean, Ph.D.
On behalf of Journal of General Physiology

Journal of General Physiology's mission is to publish mechanistic and quantitative molecular and cellular physiology of the highest quality; to provide a best-in-class author experience; and to nurture future generations of independent researchers.

Reviewer #1 (Comments to the Authors):

In this manuscript, the authors study the effect of cholesterol and cholesterol rich domains on HCN channel gating and the effect of DRG neurons of naive and injured rats. While the electrophysiological effects on HCN and DRG neurons are very convincing, the FRET-FLIM method to determine OMD size needs, in my opinion, many more controls to allow deducing information from it. The authors cite publications where the relation between FRET signal and OMD size are established, but also in those publications, it is more formulated as a suggestion and remains in very tight radii between 10 and 19Å. What is more, this was done in empty model membranes, and it is not clear what effect any membrane protein within the OMD would have on CTxB binding.

The authors state that the FRET increases upon increase in OMD diameter due to added CTxB (see points 1, 3 & 4 below), but a FRET decrease due to the larger distance would also occur (see point 3 below). The FRET distributions should be fitted directly to distributions obtained from Monte-Carlo distributions of CTxB placed in OMDs of different sizes. The resulting probability density functions should be used as weights to calculate the ensemble FRET response as shown e.g. in figure 1a-b.

It should be shown whether small OMDs with two toxins have a significant different signal from larger OMDs with three or more toxins. The principal findings described in this manuscript are probably correct, but in its current form, the data - while consistent with the conclusions

- are not sufficient to prove them. More specifically:

1. The authors should explain why the FRET signal begins at 0.05 (85Å) before increase of OMDs, if the increase is caused by addition of CTxB. The closest distance possible would be two times the radius of a toxin, which is only slightly shorter.
2. In figure 1e, the authors should elaborate which part of the figure shows that the cholesterol effect is biphasic.
3. Also in figure 1e, the authors state that the FRET-FLIM signal "plateaus" at 5 minutes. The FRET signals both do not plateau. The FRET-FLIM signal increases until 10 minutes and then slightly decreases again. This is probably due to the convoluted effects of diluting the toxins in the OMD, addition of more toxins and deformation of the OMDs by the toxins.
4. In figure 1f, the FRET efficiency is above 0.125, which would be the signal for two CTxB directly touching. Is an efficiency of 0.16 a sign of more than one acceptor being present, or do high concentrations of toxins start bending the membrane and coming closer this way? The authors should discuss whether the signal is fully saturated and therefore does not show any additional changes.
5. In figures 2c-h, the anisotropy changes antisymmetrically to the FRET signal, in accordance with the Perrin relation. The effect of the WSC in all cases is a slight increase in FRET signal. The major difference among the samples is thus the absolute FRET efficiency. The authors should discuss how reproducible the FRET efficiencies are among different preparations. How does it depend on CTxB concentration and incubation time?
6. In figure 4c the WSC-untreated SNI control is missing.
7. The authors state "consistent with a sequential cholesterol-mediated modulation of HCN function". Please elaborate how the data support this statement.
8. In the discussion of figures 4 d and g, the change in relative effect is discussed when in fact the initial value is the one that had changed. The following data points are almost identical. Please adapt the discussion.
9. Please explain the meaning of "dual modulation by OMD and cholesterol". Are not OMD rich in cholesterol?
10. Since suppl. Fig 6 is discussed for an entire paragraph, it should be included in the main manuscript.

Reviewer #2 (Comments to the Authors):

This interesting study explores the mechanism of regulation of HCN channels by cholesterol, focusing on different effects mediated by two distinct lipid pools: free cholesterol and cholesterol-rich ordered membrane domains (OMDs). The authors either added cholesterol to the plasma membrane of different cells and studied the effects of these manipulations on cholesterol distribution and HCN channel activity as a function of time. They used FRET-based cholesterol probes and advanced imaging techniques to visualize free and OMD cholesterol and found that cholesterol supplementation induced a fast expansion of OMDs, which quickly saturated, and a slower rise in free cholesterol on the inner leaflet (IL) of the plasma membrane, which allowed to temporally resolve the effects of the two cholesterol pools on HCN channel gating. Based on their electrophysiological measurements on naïve and spare nerve injury (SNI) DRG neurons, the authors proposed a model in which cholesterol modulates HCN channels in two distinct ways: by altering OMD size and directly interacting with the channels at the inner leaflet of the membrane. Some gating properties, seem to be more sensitive to OMD size while others may be more sensitive to free IL cholesterol, but the sensitivity can be changed by the pathological state of the DRG neurons (naïve vs SNI).

While the conclusions derived from the general analysis of the data are easy to understand, I found it difficult to follow the interpretation of the modulatory effect of cholesterol on the individual properties of HCN channel gating. The manuscript would benefit from a table summarizing the effects of cholesterol on gating parameters (e.g., $P_{o,max}$, $V_{1/2}$, V_s , τ_1), indicating whether each effect is more likely due to a change in OMD size or free cholesterol, with one column for naïve DRG neurons and one for SNI DRG neurons.

The effects on HCN channels observed with cholesterol removal were often the opposite of the effects observed with cholesterol supplementation. Yet, whether cholesterol removal and supplementation had opposite effects on the distribution of cholesterol between pools is unclear and needs further discussion.

The effect of cholesterol levels on membrane excitability and mechanical allodynia confirms the importance of the lipid in regulating nociceptive thresholds through the modulation of the HCN current. However, there seems to be a disconnection between this section and the mechanistic part of the study. While the changes in DRG firing rates and pain thresholds can be linked to higher or lower HCN activity, what are the relative contributions of the different cholesterol pools? and how do these contributions change in the SNI model of chronic neuropathic pain? These questions need to be addressed if the authors wish to keep the current presentation order. Alternatively, they should consider starting the manuscript with this section and then follow with the investigation of the mechanism.

Minor points:

The introduction states: "We recently uncovered that disruption of OMDs dramatically alters native HCN currents, pointing to OMDs as key modulators of HCN channel activity in the context of neuropathic pain (10, 29)." Ref. 29 is an earlier work on the effect of OMDs disruption on HCN4 activity unrelated to neuropathic pain. The sentence should be reworded to clarify the different precedents.

In Fig. 4, the time scale in panels b and c is in sec. whereas in the rest of figure it is in minutes. Keeping the units consistent across panels will improve clarity.

Reviewer #3 (Comments to the Authors):

The manuscript 'Sequential Membrane Remodeling by Cholesterol Distinctly Modulates HCN Channels in Naïve and Neuropathic DRG Neurons' by Handlin *et al.* addresses an important and experimentally challenging question: how does cholesterol modulate ion channel activity? This question is complex mainly because cholesterol can be enriched in membrane microdomains but can also exist as 'free' cholesterol in the bulk membrane. Distinguishing between these cholesterol pools within biochemically challenging membranes is difficult, creating a major gap in detailed, mechanistic understanding how cholesterol influences ion channels. The authors not only elegantly address this challenge but also go further by linking their findings to pain models, making the manuscript an intriguing piece of work linking molecular organization in the membrane with physiological response to pain.

The manuscript is well written and offers extensive solid data. All my comments are minor.

- When tsA-201 cells are first introduced, providing an explanation for choosing the cells would enhance the clarity of the results from the beginning.

- Similarly, when comparing different cell lines (small-diameter nociceptor DRG neurons from naïve rats and those subjected to the spared nerve injury (SNI)), providing more details on the rationale will help the reader understand better.

- How do the authors distinguish between current rundown/washout and cholesterol effects in their whole-cell recordings (Fig 3a-c)? A reversibility experiment involving WSC - MbCD - WSC could directly address these potential issues, specifically whether it is possible to fully recover channel function.

- "*Instead, cholesterol extraction increased the slope factor (V_s) of the G-V relationship (10), whereas WSC treatment caused only a negligible change in V_s for naïve nociceptor neurons either with or without the additional 0.5 mM cAMP (Fig. 3f).*" Cholesterol extraction is not shown in Fig 3f, and the slope factors for this treatment cannot be readily assessed. Without these data, the statement "*Since the V_s reflects voltage sensor movement and is influenced by the OMD localization of HCN channels, it could serve as a reporter of OMD dimensions of DRG neurons (10, 20). We propose that these effects arise from the naturally high cholesterol content and large OMDs in naïve DRG neurons. In this context, further increases in cholesterol levels may primarily elevate free accessible cholesterol, leading to a shift in $V_{1/2}$ and a decrease in open probability without affecting the V_s .*" is not supported.

- Page 12: for SNI nociceptor DRG neurons WSC-induced reduction in recovery is assigned a time constant of 176 s in the text, but 184 s in Fig 4c. How the reduced time constant is "*consistent with a sequential cholesterol-mediated modulation of HCN function*" is not immediately clear. This deserves more explanation.

Typos:

- Page 6: Applying [the] all four types of probes individually and overlaying their time courses

Thank you for submitting your manuscript, entitled "Sequential Membrane Remodeling by Cholesterol Distinctly Modulates HCN Channels in Naïve and Neuropathic DRG Neurons" to JGP. Your manuscript has now been seen by 3 reviewers, whose comments are appended below. You will see that all three reviewers are enthusiastic about the novelty and significance of your findings and the considerable promise of the techniques employed to evaluate membrane processes. However, they also raised several concerns that should be addressed prior to further consideration of the manuscript at JGP. In particular, please view the editor's summary directly below.

We thank the editor and reviewers for helpful advice and suggestions.

Editor Summary

Reviewers and editors are in agreement that the study addresses an important and experimentally challenging question—namely, how cholesterol modulates HCN1 channels in DRG neurons. They find the electrophysiological data compelling, the integration with pain models intriguing, and the manuscript generally well written with extensive datasets. However, several substantive issues must be addressed before the manuscript can be considered further.

1. FRET-FLIM Analysis of OMD Size: Reviewers expressed concern that the FRET-FLIM approach requires clearer justifications and additional controls. The cited literature provides only limited and model-dependent relationships between FRET signal and OMD radius, typically in simplified, protein-free membranes. The authors should:

-Add first a conceptual description of how OMD size estimation can be derived from FLIM-FRET using Alexa-tagged CTxB

We have added a conceptual description of how OMD size estimation can be derived from FLIM-FRET using Alexa-tagged CTxB. We also included new figures/modeling (Fig. 2 A-D and Supplementary Fig.1, and new method section "Poisson statistics in calculating the probability of CTxB occupancy in response to OMD expansion") based on Poisson statistics to illustrate the relationship between OMD size and the probability of multiple CTxB molecules occupying the same OMD. To provide an independent validation, we used an alternative FRET pair (L10-CFP/L10-YFP) that is membrane-anchored and independent of CTxB or GM1. We further note that this methodology has been extensively validated in our previous work (Handlin et al., 2024), which included comprehensive control experiments in native membranes and live cells.

-Provide stronger methodological justification or additional validation controls for the relation between size and FRET signal;

Now provided with new figures and simulations.

-Clarify how presence of membrane proteins in bilayer may influence CTxB binding and the OMD size interpretation;

Done

2. Support for the "Sequential Cholesterol-Mediated Modulation of HCN Function" Conclusion:

Reviewers and editors found it difficult to identify which data specifically support this central conclusion. The manuscript needs a clearer, more explicit linkage between the experimental findings and this proposed sequence of modulation. Relatedly, the assignment of time constants in the SNI DRG experiments (e.g., 176 s in text vs. 184 s in Fig. 4c) should be corrected and explained in this context.

Clarified and made the connections stronger in the current version.
Corrected. The values are consistent now.

3. Interpretation of Cholesterol Effects on HCN Gating: Although the general conclusions are understandable, the detailed interpretation of how cholesterol alters individual HCN gating properties is not sufficiently clear. The manuscript should provide a more focused and systematic explanation of how cholesterol addition versus removal affects activation, deactivation, and recovery kinetics. A table summarizing the effects of cholesterol on gating parameters (e.g., $P_{o,max}$, $V_{1/2}$, V_s , τ_1), indicating whether each effect is more likely due to a change in OMD size or free cholesterol would be beneficial. In addition, sometimes the effects on HCN channels with cholesterol removal were opposite of the effects observed when cholesterol was added. It remains unclear whether these manipulations exert opposing effects on cholesterol distribution across membrane pools. This point requires deeper discussion, ideally referencing the sequential membrane remodeling framework and how redistribution among cholesterol pools may explain differential gating effects.

We have provided a new Table S1 to summarize the detailed changes in HCN gating properties as the reviewer suggested. We also wanted to clarify that the main purpose of the paper is to develop an in vitro strategy to dissect the effects of different pools of cholesterol (the OMD pool versus the free pool) on membrane protein function.

Our previous paper (Handlin et al., 2024) studied the effects of removing cholesterol, we are not intended to fully understand the cholesterol removal on HCN gating in this paper.

4. Integration of Mechanistic and Behavioral Findings: Reviewers noted a disconnect between the mechanistic membrane/HCN experiments and the DRG excitability and mechanical allodynia data. To improve coherence, the authors should either 1) strengthen the mechanistic link by discussing the relative contributions of distinct cholesterol pools to nociceptive thresholds, including how these contributions may change in the SNI model; or 2) consider restructuring the manuscript so that the pain-behavior and DRG excitability data appear first, followed by the mechanistic investigation.

We have restructured the paper by moving the DRG excitability and mechanical allodynia data to the beginning (now Fig. 1)

5. Organization and Focus: The manuscript would benefit from improved organization and a more unified narrative. The abstract would benefit from a rewrite as it could emphasize better the central findings and conclusions of the manuscript to guide the readers through the paper. We also recommend highlighting early on and more explicitly the connection between sequential membrane remodeling with cholesterol and the differential effects of adding versus removing cholesterol on HCN gating.

Done. The abstract has been rewritten and expanded.

6. Evidence for HCN1 Association with OMDs: The evidence for HCN1 localization within OMDs should be more clearly presented. Consider including this material in the Introduction or early Results to frame the mechanistic rationale for the study.

We have included the HCN2 (main type in nociceptor DRG neurons) localization materials in the introduction.

7. Statistics: Please add exact p values to all figures.

Done.

Reviewer #1 (Comments to the Authors):

In this manuscript, the authors study the effect of cholesterol and cholesterol rich domains on HCN channel gating and the effect of DRG neurons of naive and injured rats. While the electrophysiological effects on HCN and DRG neurons are very convincing, the FRET-FLIM method to determine OMD size needs, in my opinion, many more controls to allow deducing information from it. The authors cite publications where the relation between FRET signal and OMD size are established, but also in those publications, it is more formulated as a suggestion and remains in very tight radii between 10 and 19Å. What is more, this was done in empty model membranes, and it is not clear what effect any membrane protein within the OMD would have on CTxB binding.

We thank the reviewer for these thoughtful and constructive comments. In response, we have expanded the Introduction and Methods to more clearly articulate the rationale underlying the FRET-FLIM approach used to infer changes in OMD size. We also note that the original citation list was incomplete and focused primarily on studies using model membranes. We have now included our recent work (Handlin et al., 2024), which provides extensive validation of the CTxB-based FLIM-FRET approach in native membranes and live cells, including multiple experimental controls that support its applicability beyond simplified membrane systems.

Importantly, the FRET-FLIM measurements in this study are not intended for absolute distance estimation. The donor–acceptor ratio is not 1:1, and each CTxB pentamer carries multiple fluorophores—one per protomer—connected via flexible linkers. Under these conditions, direct conversion of ensemble FRET efficiency into intermolecular distances is not meaningful. Instead, the approach is based on the principle that increasing OMD size raises the probability that two or more CTxB molecules occupy the same domain (see the new Fig. 2 A-C). The resulting increase in ensemble FRET therefore reflects changes in occupancy probability rather than changes in local intermolecular spacing.

With respect to the potential influence of membrane proteins on CTxB binding, we agree that membrane protein content could, in principle, modulate GM1 accessibility. However, this factor does not confound our interpretation because our experiments were performed in primary DRG neurons or cell lines without exogenous expression of membrane proteins, and comparisons were made within the same cellular context across conditions.

The authors state that the FRET increases upon increase in OMD diameter due to added CTxB (see points 1, 3 & 4 below), but a FRET decrease due to the larger distance would also occur (see point 3 below). The FRET distributions should be fitted directly to distributions obtained from Monte-Carlo distributions of CTxB placed in OMDs of different sizes. The resulting probability density functions should be used as weights to calculate the ensemble FRET response as shown e.g. in figure 1a-b. It should be shown whether small OMDs with two toxins have a significant different signal from larger OMDs with three or more toxins. The principal findings described in this manuscript are probably correct, but in its current form, the data - while consistent with the conclusions - are not sufficient to prove them.

We appreciate this detailed and constructive critique. In response, we have added new simulation panels to Fig. 2 that explicitly model CTxB occupancy within OMDs as a function of OMD diameter. These simulations are based on Poisson statistics evaluated along a nonlinear relationship between OMD size and available GM1 binding sites.

Specifically, we showed the probabilities of having 0, 1, 2, 3, or 4 CTxB molecules per OMD, as well as the cumulative probability of having more than two CTxB per OMD. The simulations show that this cumulative probability increases monotonically with OMD diameter, whereas the probability of only one or two CTxB molecules does not. Importantly, this increase in multi-occupancy probability directly parallels the experimentally observed increase in ensemble FRET.

These results indicate that the CTxB-based FRET signal is dominated by changes in the probability of CTxB colocalization within the same OMD, rather than by absolute intermolecular distances between individual CTxB molecules, which may indeed increase as domains expand. This behavior arises in part because CTxB binding depends on GM1, which is sparsely distributed in the plasma membrane, making multi-occupancy events highly sensitive to OMD size.

To further address this concern using an alternative approach, we employed an independent FRET reporter (L10-CFP/L10-YFP) that is membrane anchored via a short peptide, does not dissociate from the membrane, and is independent of endogenous lipids such as GM1. In this case, the observed FRET changes primarily reflect changes in intermolecular distance rather than occupancy probability. Notably, both the CTxB-based and L10-based FRET measurements report temporally similar changes in response to OMD expansion, despite relying on distinct physical mechanisms.

More specifically:

1. The authors should explain why the FRET signal begins at 0.05 (85Å) before increase of OMDs, if the increase is caused by addition of CTxB. The closest distance possible would be two times the radius of a toxin, which is only slightly shorter.

The FRET configuration used here is not intended for absolute distance estimation. Each CTxB pentamer carries multiple fluorophores (5 AF dyes) positioned at the periphery of the toxin with flexible linkers, resulting in a broad distribution of donor–acceptor distances. In addition, the measurements reflect ensemble-averaged behavior under highly heterogeneous conditions. For example, when OMDs are small, many CTxB molecules are isolated or occupy the same OMD as identical fluorophores (e.g., two donor-labeled or two acceptor-labeled CTxB), which do not contribute to FRET. These non-FRET configurations dilute the ensemble FRET efficiency.

2. In figure 1e, the authors should elaborate which part of the figure shows that the cholesterol effect is biphasic.

Clarified more in the text and figure legend.

3. Also in figure 1e, the authors state that the FRET-FLIM signal "plateaus" at 5 minutes. The FRET signals both do not plateau. The FRET-FLIM signal increases until 10 minutes and then slightly decreases again. This is probably due to the convoluted effects of diluting the toxins in the OMD, addition of more toxins and deformation of the OMDs by the toxins.

We thank the reviewer for this careful observation. Although the FRET-FLIM signal shows minor fluctuations, neither visual inspection nor statistical analysis revealed a significant difference between the signals at 10 and 15 min. Neither for the L10 FRET pairs, which are independent of GM1 density at the plasma membrane.

As supported by our Poisson-based simulations, the observed increase in ensemble FRET primarily reflects an increased probability of multiple CTxB molecules occupying the same OMD as domain size expands, rather than changes in the absolute intermolecular distance between individual CTxB molecules.

In principle, OMD expansion accompanied by a decrease in GM1 density could play a role to reduce the FRET efficiency. However, the absence of a detectable decrease after the 5 mins suggests that there is a limited sphingomyelin pool that limits the further OMD expansion. Within the first few mins, there might be a decrease in the GM1 density within OMDs, but the OMD size expansion apparently plays a predominant role, generating a net effect of increase in the FRET signal. We have added further discussion in the paper.

4. In figure 1f, the FRET efficiency is above 0.125, which would be the signal for two CTxB directly touching. Is an efficiency of 0.16 a sign of more than one acceptor being present, or do high concentrations of toxins start bending the membrane and coming closer this way? The authors should discuss whether the signal is fully saturated and therefore does not show any additional changes.

As noted above, the ensemble FRET efficiency measured here is not intended to be converted into absolute intermolecular distances. The fluorophores are attached to CTxB via flexible linkers, and each CTxB pentamer carries multiple AF dyes, resulting in a broad and heterogeneous distribution of donor–acceptor separations. Under these conditions, absolute FRET efficiencies cannot be directly mapped onto specific geometric configurations, such as two toxins in direct contact.

Rather, our interpretation of the elevated FRET efficiency (>0.125) is that it reflects an increased probability of multiple CTxB molecules co-occupying the same OMD as domain size increases, leading to enhanced heteromeric donor–acceptor interactions. This increase does not necessarily indicate reduction in intermolecular distance.

5. In figures 2c-h, the anisotropy changes antisymmetrically to the FRET signal, in accordance with the Perrin relation. The effect of the WSC in all cases is a slight increase in FRET signal. The major difference among the samples is thus the absolute FRET efficiency. The authors should discuss how reproducible the FRET efficiencies are among different preparations. How does it depend on CTxB concentration and incubation time?

We note that the anisotropy measurements were not performed using CTxB, but instead employed the cholesterol sensors GRAM-W or OlyA. In this context, anisotropy reports homo-FRET between cholesterol sensor molecules and is therefore largely independent of absolute fluorescence intensity.

The reproducibility of these measurements is reflected in the relatively small SEM values across independent experiments. In addition, because anisotropy-based homo-FRET is less sensitive to probe concentration and incubation time than hetero-FRET measurements, these data are robust across preparations.

6. In figure 4c the WSC-untreated SNI control is missing.
Added.

7. The authors state "consistent with a sequential cholesterol-mediated modulation of HCN function". Please elaborate how the data support this statement.
Modified this statement.

8. In the discussion of figures 4 d and g, the change in relative effect is discussed when in fact the initial value is the one that had changed. The following data points are almost identical. Please adapt the discussion.

Yes, that's the key difference and more discussion and a new Table S1 included. These parameters are sensitive to the initial OMD expansion.

9. Please explain the meaning of "dual modulation by OMD and cholesterol". Are not OMD rich in cholesterol?

Corrected.

10. Since suppl. Fig 6 is discussed for an entire paragraph, it should be included in the main manuscript.

We appreciate this suggestion. Supplementary Fig. 6 is discussed to provide conceptual context rather than to advance a central conclusion of the study. The sodium channel data serve a supporting role, highlighting a contrast with HCN channels: specifically, sodium channel gating exhibits a shift in $V_{1/2}$ without a change in the slope factor, whereas for HCN channels the slope factor is the primary gating parameter sensitive to OMD size rather than to changes in free cholesterol.

Reviewer #2 (Comments to the Authors):

This interesting study explores the mechanism of regulation of HCN channels by cholesterol, focusing on different effects mediated by two distinct lipid pools: free cholesterol and cholesterol-rich ordered membrane domains (OMDs). The authors either added cholesterol to the plasma membrane of different cells and studied the effects of these manipulations on cholesterol distribution and HCN channel activity as a function of time. They used FRET-based cholesterol probes and advanced imaging techniques to visualize free and OMD cholesterol and found that cholesterol supplementation induced a fast expansion of OMDs, which quickly saturated, and a slower rise in free cholesterol on the inner leaflet (IL) of the plasma membrane, which allowed to temporally resolve the effects of the two cholesterol pools on HCN channel gating. Based on their electrophysiological measurements on naïve and spare nerve injury (SNI) DRG neurons, the authors proposed a model in which cholesterol modulates HCN channels in two distinct ways: by altering OMD size and directly interacting with the channels at the inner leaflet of the membrane. Some gating properties, seem to be more sensitive to OMD size while others may be more sensitive to free IL cholesterol, but the sensitivity can be changed by the pathological state of the DRG neurons (naïve vs SNI).

While the conclusions derived from the general analysis of the data are easy to understand, I found it difficult to follow the interpretation of the modulatory effect of cholesterol on the individual properties of HCN channel gating. The manuscript would benefit from a table summarizing the effects of cholesterol on gating parameters (e.g., $P_{o,max}$, $V_{1/2}$, V_s , τ_1), indicating whether each effect is more likely due to a change in OMD size or free cholesterol, with one column for naïve DRG neurons and one for SNI DRG neurons.

Thanks for this suggestion, we have induced a new Table S1

The effects on HCN channels observed with cholesterol removal were often the opposite of the effects observed with cholesterol supplementation. Yet, whether cholesterol removal and supplementation had opposite effects on the distribution of cholesterol between pools is unclear and needs further discussion.

We thank the reviewer for this comment. Cholesterol removal was extensively studied in our previous work (Handlin et al., 2024). In the present study, our focus is on methodological development and on understanding how distinct plasma membrane cholesterol pools influence HCN channel gating. We intentionally did not include cholesterol removal experiments here, as this treatment is relatively harsh and can trigger signaling effects that could confound the interpretation of how specific cholesterol pools modulate HCN channels.

The effect of cholesterol levels on membrane excitability and mechanical allodynia confirms the importance of the lipid in regulating nociceptive thresholds through the modulation of the HCN current. However, there seems to be a disconnection between this section and the mechanistic part of the study. While the changes in DRG firing rates and pain thresholds can be linked to higher or lower HCN activity, what are the relative contributions of the different cholesterol pools? and how do these contributions change in the SNI model of chronic neuropathic pain? These questions need to be addressed if the authors wish to keep the current presentation order. Alternatively, they should consider starting the manuscript with this section and then follow with the investigation of the mechanism.

We decided to take the reviewer's suggestion and moved this part to the beginning of the paper.

Minor points:

The introduction states: "We recently uncovered that disruption of OMDs dramatically alters native HCN currents, pointing to OMDs as key modulators of HCN channel activity in the context of neuropathic pain (10, 29)." Ref. 29 is an earlier work on the effect of OMDs disruption on HCN4 activity unrelated to neuropathic pain. The sentence should be reworded to clarify the different precedents.

Corrected, removed one of the references here.

In Fig. 4, the time scale in panels b and c is in sec. whereas in the rest of figure it is in minutes. Keeping the units consistent across panels will improve clarity.

Changed.

Reviewer #3 (Comments to the Authors):

The manuscript 'Sequential Membrane Remodeling by Cholesterol Distinctly Modulates HCN Channels in Naïve and Neuropathic DRG Neurons' by Handlin *et al.* addresses an important and experimentally challenging question: how does cholesterol modulate ion channel activity? This question is complex mainly because cholesterol can be enriched in membrane microdomains but can also exist as 'free' cholesterol in the bulk membrane. Distinguishing between these cholesterol pools within biochemically challenging membranes is difficult, creating a major gap in detailed, mechanistic understanding how cholesterol influences ion channels. The authors not only elegantly address this challenge but also go further by linking their findings to pain models, making the manuscript an intriguing piece of work linking molecular organization in the membrane with physiological response to pain.

We thank the reviewer's positive comments.

The manuscript is well written and offers extensive solid data. All my comments are minor.

- When tsA-201 cells are first introduced, providing an explanation for choosing the cells would enhance the clarity of the results from the beginning.

Added.

- Similarly, when comparing different cell lines (small-diameter nociceptor DRG neurons from naïve rats and those subjected to the spared nerve injury (SNI)), providing more details on the rationale will help the reader understand better.

Added.

- How do the authors distinguish between current rundown/washout and cholesterol effects in their whole-cell recordings (Fig 3a-c)? A reversibility experiment involving WSC - MbCD - WSC could directly address these potential issues, specifically whether it is possible to fully recover channel function.

The rundown was studied using no treatment controls as shown in the supplementary Figure 6. The longer duration experiments of sequential treatments are plausible, but experimentally difficult since patches typically last only 15-20 mins.

"Instead, cholesterol extraction increased the slope factor (V_s) of the G-V relationship (10), whereas WSC treatment caused only a negligible change in V_s for naïve nociceptor neurons either with or without the additional 0.5 mM cAMP (Fig. 3f)." Cholesterol extraction is not shown in Fig 3f, and the slope factors for this treatment cannot be readily assessed. Without these data, the statement "Since the V_s reflects voltage sensor movement and is influenced by the OMD localization of HCN channels, it could serve as a reporter of OMD dimensions of DRG neurons (10, 20). We propose that these effects arise from the naturally high cholesterol content and large OMDs in naïve DRG neurons. In this context, further increases in cholesterol levels may primarily elevate free accessible cholesterol, leading to a shift in $V_{1/2}$ and a decrease in open probability without affecting the V_s ." is not supported.

This effect by cholesterol extraction was not shown because it was published in our previous paper (Handlin et al., 2024). We have further emphasized the published cholesterol extraction result in the text.

This current paper was an extension of the previous paper.

- Page 12: for SNI nociceptor DRG neurons WSC-induced reduction in recovery is assigned a time constant of 176 s in the text, but 184 s in Fig 4c. How the reduced time constant is "*consistent with a sequential cholesterol-mediated modulation of HCN function*" is not immediately clear. This deserves more explanation.

Corrected the value. It should be 184 s. We have modified this statement.

Typos:

- Page 6: Applying [the] all four types of probes individually and overlaying their time courses

Corrected.

Dr. Gucan Dai
Saint Louis University
Department of Biochemistry and Molecular Biology
School of Medicine
St. Louis, MO 63104

Re: 202513925R1

Dear Gucan,

I am pleased to let you know that your manuscript, titled "Sequential Membrane Remodeling by Cholesterol Distinctly Modulates HCN Channels in Naïve and Neuropathic DRG Neurons" is scientifically acceptable for publication in Journal of General Physiology. Formal acceptance will follow when it is modified in accordance with the referees' remarks and our editorial policies.

The editors have a few minor requests. First, please provide exact p values for all statistical comparisons. While these are reported in most cases, some instances still indicate $p < \dots$ (e.g. Fig S6 and S7, and elsewhere). Please revise these to include the precise values. Second, please include in the Methods or Results section, a detailed description of the protocol used for cholesterol manipulation for both the electrophysiology and fluorescence experiments. This should specify incubation time with β -cyclodextrin and/or cholesterol as well as any relevant experimental details necessary for reproducibility.

Please note other items that need attention are listed at the bottom of this email (under 'manuscript formatting checklist'). Please also be sure to include a letter addressing the reviewers' comments point-by-point (if applicable) and a copy of the text with alterations highlighted (boldfaced or underlined). Your manuscript should be a double-spaced MS Word file and include editable tables, if appropriate.

Lastly, JGP requires a data availability statement for all research article submissions. These statements will be published in the article directly above the Acknowledgments. The statement should address all data underlying the research presented in the manuscript. Please visit the JGP instructions for authors for guidelines and examples of statements at <https://rupress.org/jgp/pages/editorial-policies#data-availability-statement>.

Please submit your final files via this link:

Link Not Available

Thank you for choosing to publish your research in JGP and please feel free to contact me with any questions.

Sincerely,

Crina Nimigean, Ph.D.
On behalf of Journal of General Physiology

Journal of General Physiology's mission is to publish mechanistic and quantitative molecular and cellular physiology of the highest quality; to provide a best in class author experience; and to nurture future generations of independent researchers.

Manuscript formatting checklist:

- MS Word document of text needed (including editable tables)
- MS Word document of supplemental text needed, if applicable (including figure legends and editable tables)
- Brief Statement describing supplementary information needed, if applicable (in subsection at end of Materials & Methods)
- Please include a data availability statement preceding the Acknowledgments section. Please see <https://rupress.org/jgp/pages/editorial-policies#data-availability-statement>
- Figures created at sufficient resolution and in acceptable format (including supplemental if applicable). If working in Illustrator, we prefer .ai or .eps file format. If working in Photoshop please use 600dpi/1000dpi .tiff or .psd file format. Minimum resolution at estimated print size: Minimum resolution for all figures is 600 dpi. For figures that contain both photographs and line art or text, 600 dpi is highly recommended. Figures containing only black and white elements (line art, no color, and no gray) should be 1,000 dpi. Maximum figure size is 7 in wide x 9 in high (17.5 x 22.8 cm) at the correct resolution. <https://jgp.rupress.org/fig-vid-guidelines>
- Supplemental figures, if any, conforming to same guidelines as manuscript figures (noted above)
- If images resemble one from a prior publications, the author must seek permissions (to reproduce or adapt) from the original publisher. [You can resubmit your paper while waiting to hear back from the original publisher but please keep us updated]
- All authors must complete a disclosure form prior to acceptance. A link to complete the form has been sent to all coauthors.

Please provide the editorial office with updated email addresses if necessary

Reviewer #1 (Comments to the Authors):

The authors responded very well to all reviewers' comments.

Reviewer #2 (Comments to the Authors):

I am satisfied with the authors' revisions. I have no further concerns.

Reviewer #3 (Comments to the Authors):

All my comments have been addressed.